# Drug Use in Street Sex worKers (DUSSK) study: results of a mixed methods feasibility study of a complex intervention to reduce illicit drug use in drug dependent female sex workers

Rita Patel [1,2] Niamh M Redmond [1,2] Joanna M Kesten,[1,3] Myles-Jay Linton,[1,4] Jeremy Horwood,[1,5] David Wilcox,[6] Jess Munafo,[6] Joanna Coast,[1,4] John Macleod,[1,5] Nicola Jeal[1,2,7]

RP and NMR are joint first authors.

For numbered affiliations see end of article.

**Correspondence to**
Dr Nicola Jeal;
Nikki.Jeal1@nhs.net

## ABSTRACT

**Objectives** The majority of female street-based sex workers (SSWs) are dependent on illicit drugs and sell sex to fund their drug use. They typically face multiple traumatic experiences, starting at a young age, which continue through sex work involvement. Their trauma-related symptoms tend to increase when drug use is reduced, hindering sustained reduction. Providing specialist trauma care to address post-traumatic stress disorder (PTSD) alongside drug treatment may therefore improve treatment outcomes. Aims to (1) evaluate recruitment and retention of participants; (2) examine intervention experiences and acceptability; and (3) explore intervention costs using a mixed methods feasibility study.

**Setting** Female SSW charity premises in a large UK inner city.

**Participants** Females aged 18 years or older, who have sold sex on the street and used heroin and/or crack cocaine at least once a week in the last calendar month.

**Intervention** Female SSW-only drug treatment groups in a female SSW-only setting delivered by female staff. Targeted PTSD screening then treatment of positive diagnoses with eye movement desensitisation and reprocessing (EMDR) therapy by female staff from a specialist National Health Service trauma service.

**Results** (1) Of 125 contacts, 11 met inclusion criteria and provided informed consent, 4 reached the intervention final stage, (2) service providers said working in collaboration with other services was valuable, the intervention was worthwhile and had a positive influence on participants. Participants viewed recruitment as acceptable and experienced the intervention positively. The unsettled nature of participant's lives was a key attendance barrier. (3) The total cost of the intervention was £11 710, with staff costs dominating.

**Conclusions** Recruitment and retention rates reflected study inclusion criteria targeting women with the most complex needs. Two participants received EMDR demonstrating that the three agencies working together was feasible. Staff heavy costs highlight the importance of supporting participant attendance to minimise per participant costs in a future trial.

## Strengths and limitations of this study

► The novel intervention integrates a trauma-focussed treatment approach in order to reduce drug use in a challenging drug treatment population.
► The intervention was delivered by specialists, reflecting the skills and experience required to appropriately manage the complex needs of the study population.
► Patient and public involvement formed an important part of the study methodology and informed each stage.
► Recruitment took place over several months within an agency trusted and used daily by the study population to allow familiarisation with the researchers and multiple opportunities to participate.
► This feasibility study design and methodology was not able to examine intervention effectiveness or cost-effectiveness.

## INTRODUCTION

Most female street-based sex workers (SSWs) in the UK use heroin and/or crack cocaine.[1–3] Their drug dependency keeps them entrenched in a 'work-score-use cycle',[4 5] which contributes to the morbidity and social instability typically seen in this group.[6]

Despite their drug treatment needs, drug dependent SSWs have poorer outcomes from drug treatment services compared with other service users,[7 8] sometimes due to stigma associated with their street sex work.[9] Previous SSW-focussed interventions aiming to reduce drug use have used educational,[10 11] substitute prescribing-based[12 13] and psychological[14] approaches but none robustly demonstrated a positive effect in reducing drug use.[15]

Poor mental health is a significant problem among SSWs.[16–18] Many have experienced

multiple adversities in early life and during their involvement in sex work,[5 16 19] which exposes women to further risk of significant trauma.[16 17] Consequently, many SSWs are affected by post-traumatic stress disorder (PTSD).[16 17] Trauma symptoms, which often recur when drug use is reduced, may motivate a return to drug use.[20] Individual trauma-focused therapy alongside drug treatment may provide the best outcomes for reductions in drug use.[21–23] However to date, there is no robust evidence to demonstrate the impact of an integrated trauma-focussed treatment approach in reducing drug use among female drug dependent SSWs.

In collaboration with SSWs and service providers, and informed by existing research,[9 15] we developed a novel intervention, to simultaneously address the unique and complex combination of drug use and PTSD in female drug-dependent SSWs. The intervention proposes an integrated care pathway through an innovative multi-agency partnership.[24] We report here the results of the Drug Use in Street Sex worKers (DUSSK) feasibility study, which aimed to (1) evaluate the recruitment and retention of SSWs to the intervention; (2) examine the experience and acceptability of the intervention for participants and service providers; and (3) explore costs to service providers associated with the intervention.

## METHODS

### Study design, setting and eligibility

Detailed methods are described in the published protocol.[24] This mixed methods feasibility study took place in a UK inner city setting. Females aged 18 years or older, who sold sex on the street at least weekly in the last calendar month and used heroin and/or crack cocaine at least once a week in the last calendar month, were eligible to participate.[24] The intervention was delivered at SSW charity premises, which supplied support, health and advocacy services.[24]

### Recruitment

The recruitment target for this feasibility study was 30 women.[24] Local organisations that SSWs were known to access were provided with study promotional materials. One of three researchers (JMK, NJ, SR) attended an SSW support and advocacy charity, at least twice a week, to directly approach potential participants; alternatively, interested SSWs could telephone researchers. This approach meant that women were potentially approached and counted as contacts several times during recruitment. Eligibility screening was conducted face-to-face or via telephone. Women gave fully informed, written consent to participate in the study and baseline data were also collected at the time of recruitment. To maintain safety and confidentiality, each participant provided details of acceptable ways in which to be contacted. Screening data were retained and remained anonymised for those not recruited.

### Patient and public involvement (PPI)

Women with experience of street sex work and drug-dependency took part in focus groups and one-to-one discussions with NJ to inform study design, processes, documentation and intervention development. On commencement of the study, a group of women who were ineligible for recruitment were approached for involvement in PPI. They addressed challenges with recruitment, participation and adherence issues (described below) and suggested solutions, which were implemented. For example, they recommended changes such as provision of sandwiches to improve attendance.

### The intervention

The intervention consisted of SSW only drug treatment groups, targeted screening for PTSD symptoms (one-to-one clinical interview and PTSD Checklist (PCL5))[25] and, if positively diagnosed, one-to-one eye movement desensitisation and reprocessing (EMDR) therapy, all delivered by female staff through a collaboration between three service providers (National Health Service (NHS) trauma services, the SSW and drug treatment charities). The intervention was designed so participants were initially invited to attend a weekly 'Getting started' drug treatment group to reduce fear and anxiety about engaging in a group setting and get used to the format and level of disclosure expected. Participants were to progress to a 'Preparation for recovery' drug treatment group which focused on people's barriers to motivation for change, examining pros and cons of drug use and exploring triggers for using drugs to enable participants to achieve a level of drug use stability. As stated in the protocol[24] and in line with provider's usual care, participants were perceived as demonstrating drug use stability by exhibiting evidence of life/drug use stability such as engagement and functioning in the group, positive interaction with group facilitators and regular opioid substitution therapy (OST) by the group facilitators. When group facilitators judged participants were achieving drug use stabilisation, and they had attended three sessions consecutively, they were offered screening for PTSD symptoms by a female clinical psychologist. Those experiencing PTSD symptoms were invited to attend five PTSD 'Stabilisation' group sessions, facilitated by the same female clinical psychologist, to equip participants with the skills to self-soothe and reorientate in preparation for the one-to-one EMDR treatment. Once all 'Stabilisation' group sessions had been completed, the clinical psychologist assessed participants for readiness for one-to-one EMDR sessions and if eligible, participants progressed to a course of 12 sessions with the clinical psychologist on a weekly, or fortnightly, basis. Trauma treatment (PTSD screening, stabilisation groups and one-to-one) ran in parallel to the drug treatment groups; figure 1 (red boxes) shows the planned flow of participants through the study.

The intervention proceeded as described in the protocol paper[24] with the following changes (bulleted next):

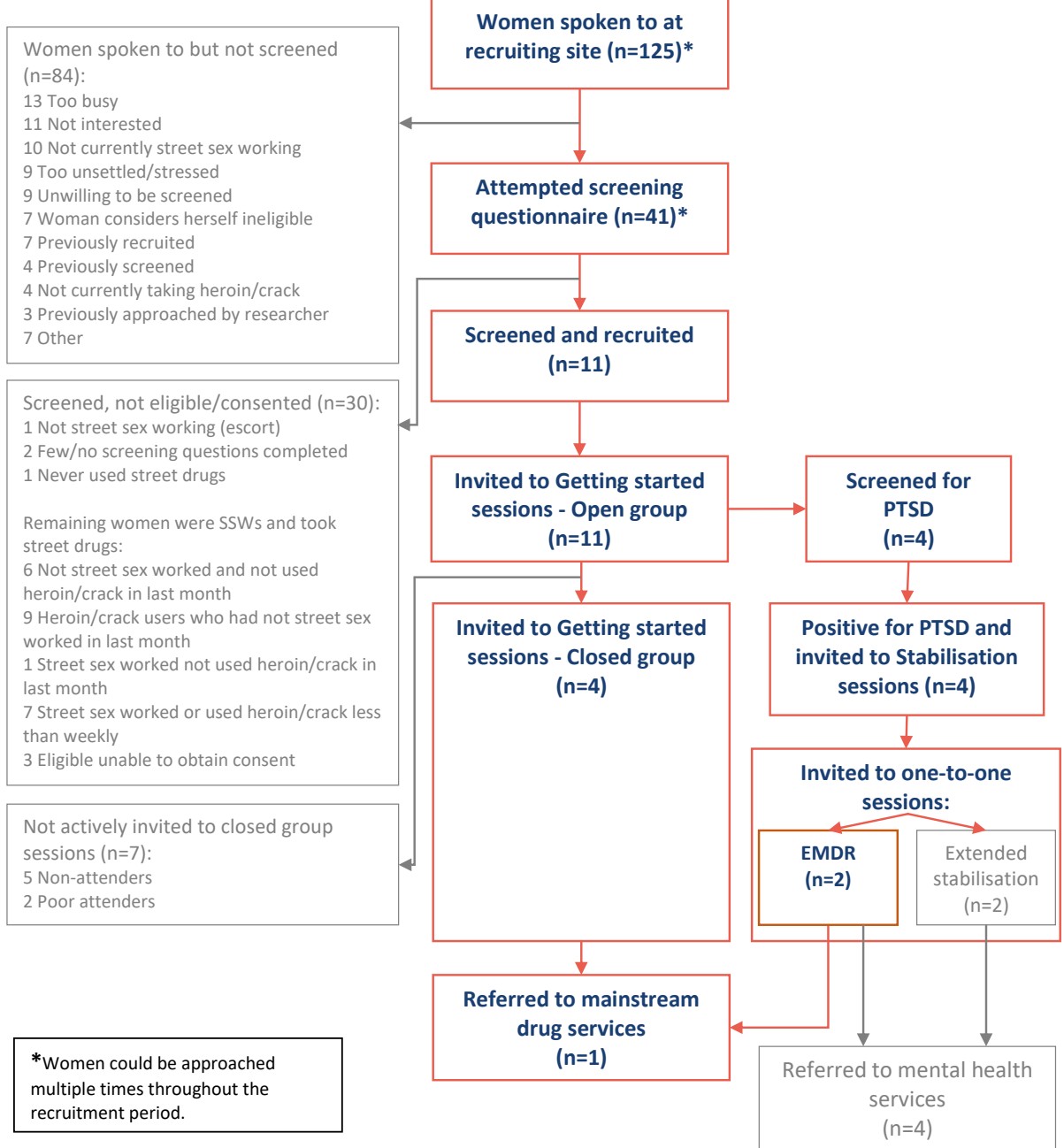

**Figure 1** Flow of participants through the Drug Use in Street Sex worKers study. PTSD, post-traumatic stress disorder; SSWs, street-based sex workers; EMDR, eye movement desensitisation and reprocessing therapy

► Participants were encouraged to attend all sessions with the offer of car lifts, bus tickets and taxis, in addition to the planned weekly phone calls and texts, by the staff at partner agencies.

### 'Getting started' and 'Preparation for recovery' drug treatment groups

► Retendering resulted in a change of drug group service provider which, along with low numbers recruited, resulted in the drug groups merging into a single open drug treatment 'Getting started' group.

► Attendance at three consecutive sessions was required to move onto the next group, instead of attendance at any four.

► Sandwiches were supplied prior to the single drug group to support attendance from the 14th session onwards.

- The number of sessions continued beyond those initially planned due to delays in PTSD screening (see below).
- Poor attendance, the participant's unstable behaviour and the intervention running for longer than planned (due to the retendering of drug services) affected the drug group facilitators' ability to deliver structured content. An art worker was included in four sessions to maintain participant's interest and engagement with sessions.
- PTSD screening was halted from the 29th session onwards, due to the limited remaining study time to complete the intervention, meaning the drug group became closed and only included PTSD screened participants.

### Screening for PTSD

- Delays occurred to both the organisation of the PTSD screening and PTSD stabilisation group set up due to lack of capacity and delays in NHS trauma service staff recruitment. Therefore, some participants had long gaps between recruitment and PTSD screening and subsequent stabilisation group sessions.

### PTSD 'Stabilisation' group

- Incentives of £10 vouchers per session were offered for attendance (mandatory sessions).

### One-to-one EMDR therapy for PTSD

- Some sessions were scheduled in a private rented room (due to availability issues) in a local community centre and not at the SSW charity premises.
- Two participants were offered weekly one-to-one extended stabilisation sessions (eight maximum) with the clinical psychologist rather than EMDR therapy as this was deemed the most appropriate treatment.

### Data collection methods

#### Sample size and quantitative data collection

A formal sample size calculation was not conducted as the aim was to assess feasibility.[24] Self-reported levels of illicit drug use, involvement in SSW, completion of PTSD Checklist PCL5[25] and demographics were collected at the time of consent. Attendance registers were taken at the start of each group or one-to-one session by the facilitator(s).

#### Qualitative data collection

With participants' verbal consent one non-participant observation of a drug group was conducted to understand delivery, examine interactions and intervention experiences, with brief notes taken during the group.[26]

In-depth semistructured interviews were conducted with participants and service providers either face-to-face or by telephone. Participants were interviewed after intervention completion or study drop out. Consent to contact participants regarding interviews was sought at recruitment. Additional written or audio recorded verbal, informed consent was obtained prior to all interviews.

Interviews explored views and experiences of the intervention and how to improve acceptability. Participants received a £20 high street shopping voucher for taking part. Most service provider interviews were conducted at the end of the intervention period and also sought to understand operational issues, interagency working and intervention delivery.

#### Economic data collection

Resource use information (2018 £GBP) was collected prospectively by the agencies. Data collection focused on four categories: staff time, facilities, travel (provider funded staff and patient transport) and materials.

Non-attendance was dealt with as follows; if no participants arrived after 45 min for a 'getting started' session, staff left and were only costed for the time that they spent waiting. Staff delivering 'trauma screening', '1–1 s' and 'stabilisation groups' were costed for all sessions booked, regardless of non-attendances.

### Data analysis

The integration of qualitative and quantitative data used the established 'following a thread' technique[27] where key themes were traced using all data sets.

#### Statistical analysis

Descriptive statistics (using Stata v.14) were used to monitor recruitment and retention via CONSORT statement[28] style flow charts, and to examine participant demographics, questionnaire responses and attendance.

#### Qualitative analysis

Interviews were conducted by JMK and NJ, and the audio files transcribed, anonymised and checked for accuracy. QSR NVivo v.10 software was used to perform inductive thematic analysis[29] using constant comparison techniques.[30 31] A preliminary coding framework was developed by JMK and discussed with the multidisciplinary research team, JH and NJ to ensure credibility and confirmability.

#### Cost analysis

Costs to the service provider were examined and summarised separately for each of the four intervention components. Staff costs were calculated using salaries and oncosts or generated using standard unit cost data available for health and social care professionals.[32] Facility costs were calculated based on similar space rental options. Total cost, total cost per eligible participant, total cost per session held, and total cost per session per eligible participant were calculated for each intervention component. Cost data were tabulated using principles of heat-map methodology, where colour lightness is used to communicate the magnitude of different costs.[33 34]

### RESULTS

Recruitment was from November 2017 to March 2018 with the intervention delivered until December 2018.

## Recruitment and retention
### Recruitment process

Approximately 400 flyers and nine posters were distributed. Potential participants were spoken to by three researchers at 37 3 hour 'drop-in' sessions at the SSW charity. Of 125 contacts made with women, 84 declined screening. Reasons for decline included being too busy (n=13), not interested (n=11) or reporting not currently street sex working (n=10). Fourteen approaches were reported as repeat approaches, with seven reporting previous recruitment and four previous screening (as the researchers recruited on different weekdays and screening data was anonymous). Figure 1 details the flow of screened and recruited participants through the intervention.

Of 41 women screened, 11 were eligible and consented to participate, 3 were eligible but unable to give consent (2 were too distressed and 1 had health issues preventing participation). Of the 27 ineligible women, two did not fully complete the screening questions. The main reasons for exclusion related to ineligible frequency of drug use and/or sex work. Table 1 shows the range of days since responders last street sex worked or had taken heroin/crack by those recruited and not recruited.

Of 11 participants consenting to be invited for qualitative interviews, 5 were uncontactable. Seven interviews were conducted with six participants (six face-to-face and one via telephone); four interviewed participants received all components of the intervention. Ten service provider interviews were conducted with representatives from the drug treatment service (n=4), the trauma service (n=2), and the SSW charity (n=4). Table 2 details quotes from the interviews to support the main results.

### Acceptability of recruitment

The recruitment process was described as acceptable by most participants and service providers. Face-to-face recruitment was experienced as confidential, with participants reporting receiving clear explanations of the study. Recruitment over 5 months within the SSW charity offered multiple opportunities to participate and was acceptable. Most service providers interviewed reflected that if study inclusion criteria were broadened to include less regular drug use, participants with more lifestyle stability could have been recruited, and whom may have found the intervention easier to engage with. However, some felt that creating groups with diverse levels of drug use could have negative consequences for example, risk of relapse for those who had reduced their drug use.

### Group attendance

All 11 consented participants were invited to attend drug treatment groups. However, participants attendance varied throughout the study, with participants sometimes arriving late or leaving early (table 3). Four attended 30%–76% of sessions compared with 7 attenders who attended only 0%–18% of sessions. The five most frequent attenders were invited to PTSD screening of which four attended and were all found to have symptoms of PTSD. All four PTSD screened participants attended 20%–100% of the stabilisation groups with the clinical psychologist. At the end of the stabilisation groups, two participants were deemed suitable for EMDR therapy by the clinical psychologist and two were offered extended stabilisation sessions. All four participants attended some one-to-one sessions (table 3) but missed at least two consecutive trauma treatment appointments with the clinical psychologist and had to withdraw from the sessions. However, all participants were referred to further mental health services and one participant was also referred to mainstream drug services.

### Facilitators to attendance

Service providers across all partner agencies sent reminders to participants, which were described as helpful and appreciated by participants. One SSW charity service provider played a vital role in encouraging attendance through reminding participants to attend, arranging transport (taxi, bus or driving participants to sessions) and helping participants prepare for the intervention. Provision of sandwich lunches before the groups was seen by service providers and participants as helpful for encouraging attendance, facilitating a relaxed start to groups and supporting concentration. Vouchers were also viewed as encouraging attendance by participants and service providers.

### Barriers to attendance

The unsettled nature of participant's lives was perceived as an attendance barrier and was underpinned by problematic drug use, poor adherence to OST, sex work, tiredness and poor mental health. Arguments between participants, a lack of readiness to confront issues with drugs and trauma and an absence of social support were described as making attendance difficult. Delays to screening referral and trauma treatment were also reported to negatively affect participants' motivation.

## Experience and acceptability of the intervention
### Initial impressions

All participants perceived the intervention as valuable and welcome. Common reasons given by participants for taking part included the opportunity for 'change', greater stability and valuing the opportunity to combine mental health and drug treatment. Service providers viewed the intervention as a novel opportunity for (1) SSWs to receive mental health treatment while continuing to use drugs and (2) to address the barriers to mainstream drug treatment. Some service providers highlighted the challenge for participants to process trauma while continuing to use drugs and potential risks of sex working when receiving trauma treatment.

### Service providers' views on the intervention

The drug group facilitators described enjoying delivering the groups and building good relationships with participants. However, they described some drug sessions

**Table 1** Characteristics of screened women

| | Screened not recruited (N=30) | | Screened and recruited (N=11) | |
|---|---|---|---|---|
| | N (%)* | Median (range) | N (%)* | Median (range) |
| Female | 27 (90) | | 11 (100) | |
| Age | 23 (77) | 37 (26–55) | 11 (100) | 38 (23–53) |
| Ever sold sex on the street? | | | | |
| Yes | 26 (87) | | 11 (100) | |
| No | 1 (3) | | – | |
| How many days since last worked on the street? | 23 (77) | 60 (1–2920) | 11 (100) | 7 (1–28) |
| How often usually sell sex on street? | | | | |
| Daily | 6 (20) | | 3 (27) | |
| Weekly | 5 (17) | | 8 (73) | |
| Less than weekly | 16 (53) | | – | |
| Ever used street drugs | | | | |
| Yes | 26 (87) | | 11 (100) | |
| No | 1 (3) | | – | |
| Ever used heroin | 23 (77) | | 9 (82) | |
| Days since last used heroin | 19 (63) | 2 (0–731) | 9 (82) | 1 (0–6) |
| Ever used crack cocaine | 23 (77) | | 11(100) | |
| Days since last used crack cocaine | 21 (70) | 2 (0–2922) | | 1 (0–4) |
| How often use heroin and/or crack cocaine? | | | | |
| Daily | 11 (37) | | 7 (64) | |
| Weekly | 4 (13) | | 4 (36) | |
| Less than weekly | 9 (30) | | – | |
| Has an opioid substitute script | 13 (43) | | 6 (55) | |
| Script type | | | | |
| Buprenorphine/Subtex | – | | 1 (9) | |
| Methadone | 13 (43) | | 5 (45) | |
| Used other drugs: | | | | |
| Alcohol | 3 (10) | | 1 (9) | |
| Amphetamine | 1 (3) | | – | |
| Cannabis | 5 (17) | | 5 (45) | |
| Spice | 2 (7) | | – | |
| MDMA (Ecstasy) | 1 (3) | | | |
| Tramadol | 1 (3) | | – | |
| Sleeping tablets | – | | 1 (9) | |
| PCL5 score (possible range 0–80) | – | | 10 (91) | 56 (43–73) |

MDMA, Methylenedioxymethamphetamine
*N and % of those that provided data

as intense and difficult to manage due to participants' distress, accounts of trauma and chaotic behaviour. The need for appropriate support and supervision of facilitators was highlighted as a requirement to manage these challenges.

A clinical psychologist suggested that without the 're-traumatising' effects of street sex work, the effectiveness of the trauma processing in the trauma treatment might be enhanced. Service providers also proposed extending the stabilisation work to develop the effectiveness of the trauma treatment and recommended the intervention offer alternatives to EMDR to suit individual participants' needs.

Service providers said working in partnership with other specialist services to deliver the intervention was valuable, there was mutual respect and good communication

**Table 2** Qualitative quotes

| Theme/subtheme | Quotes |
|---|---|
| **Recruitment and retention** | |
| Acceptability | *(Face-to-face recruitment)worked well, it wasn't intrusive, you weren't pushy, you know you blended in within the drop-in setting. So I think the women felt that if they did wanna buy into it they would, there was no pressure there. So I think that was done really sensitively*. Service provider 6 |
| | *It [recruitment] was very sort of like confidential and actually it was quite nice 'cause, yeah no one really knew what I was doing when I was doing summut, you know what I mean, which is – like it don't usually happen like that. Everyone knows what I'm doing all the time*. Participant 7 |
| Improvements | *I think from a clinical point of view if you remove that criteria (sex work at least once a week in the last calendar month) and then of course there's more chance of getting people through to the finish line to be able to be ready for treatment at the end*. Service provider 4 |
| | *Actively drug using? Yes, that makes sense (…). If they've been able to bring that down themselves maybe another service would be better. Like, what this offered, it's specialistic in this. So if you was able to manage to a level yourself, maybe you don't need [the intervention]… I'm not sure, I think that would be an interesting conversation because if they could bring it down themselves, they'd probably be a lot more stable and a lot more reliable to actually get to the EMDR* . Service provider 7 |
| | *So I think if you were to extend the period of time and say 'Oh actually do you know if you've used within the last three months you can participate in the study and then someone who's three months abstinent or reducing from their street heroin use or their crack use is then exposed to somebody who's going no no no man I'm using up like a party every night'. There'd be that ethical thing within it but it would be nice to see the study opened up to a wider cohort*. Service provider 2 |
| **Facilitators to attendance** | |
| Encouragement and support to attend | *I would say that I've been quite integral in regards to developing relationships with the women, contacting them for both their individual one to ones and stabilisation groups and also their Thursday DUSSK groups as well. So just keeping that contact going if they were coming in, in our drop-in service I would see them and then sort of give them reminders, did they want little welfare calls, that type of thing*. Service provider 6 |
| Transport | *It was more of a focus thing where you know she(service provider 6)sort of like coached us as we went down, like you know keeping us like sort of aware of what we've got to be thinking of doing and making sure that, you know, there's nothing wrong*. Participant 7 |
| Food provision | *I was turning up and I was like sort of god like hanging out for (…) that lunch. It was like, not the reason I was turning up but the main reason why I could (…). There is light at the end of the tunnel, you know you're gonna be nourished and fed.' You're gonna be able to concentrate as well*. Participant 7 |
| **Barriers to attendance** | |
| Unstable lifestyles | *My mental problems, my drug use, everything, just my life, it gets in the way [of attendance]*. Participant 1 |
| Mental health | *They're so low resourced, they just don't have the distress tolerance to be able to cope with any more distress, they're already facing so much. Even things like their housing and threats of eviction*. Service provider 8 |
| | *My home life was getting a bit chaotic. My depression was getting really bad as well. So, yeah, and I was waiting for my antidepressants to work but they took a while. Yeah, it was just my depression, that's all*. My anxiety. Participant 6 |
| Sex work | *If I've been working the night before there's no way I could have attended because I'm too tired, because you work all night*. Participant 4 |
| Delays between treatment stages | *It took a little bit of a while and also for them to access their stabilisation groups then their one to ones. I think we may have lost some of the interest*. Service provider 6 |
| **Experience and acceptability of the intervention** | |
| Initial impressions | *There aren't many services out there, which will offer individual, tailorised counselling and support to the women who have got dual diagnosis and you know mental health, drug misuse. So this study was unique and I think that's what we were all so passionate and so behind it because it was giving the women an opportunity*. Service provider 6 |

Continued

**Table 2** Continued

| Theme/subtheme | Quotes |
|---|---|
| Reason for participating | *I just felt so alone and afraid and stuck and just needed to see if there was some way that I might be able to gain something so—really, if I'm willing to put myself out on the street and sell myself to a complete stranger, knowing that I might die, whatever, so it kind of … I felt I needed to understand why I needed to do this… So it's about me owning my power, and about not letting myself feel as shit about myself as I have done.* Participant 5<br>*The post-traumatic stress [treatment] is—is a way of like sort of detoxing your brain. So, you know finding a reason why you do these drugs (…) to like sort of be the reason for me to like say 'Well, I've got to stop now.' You know and get off it.* Participant 7 |
| Service providers views on the intervention | *I guess that people thought they weren't going to talk about their traumas [in the drug groups] but if somebody's been raped last night, they're going to need to talk about it, so we were here dealing with that stuff on the spot and then we didn't have no-one to go away and talk about it.* Service provider 10<br>*It's very hard to do trauma processing when women to some degree are being traumatised and then having to self-medicate against all of that and then you're trying to work on quite deep attachment developmental trauma stuff from a long time ago. (…) I'd say that trauma processing would be more successful with women who have maybe made a very strong commitment to stop [sex] working.* Service provider 8<br>*I would offer it [EMDR] as part of a—as a range of things that are offered…we'd say 'You can have EMDR, trauma focus CBT [Cognitive Behavioural Therapy] or narrative exposure therapy and you'd kind of match the person to what you thought they might be more suited to.* Service provider 8<br>*I think they had huge admiration for the workers at [SSW charity], and found them friendly and supportive, but…there wasn't a specific, I don't know, once a month structured 'let's talk about the women and how they've been in the month.* Service provider 9 |
| Participants views of the intervention | *I enjoyed going down there. (…) We had a good laugh and learned something while we were doing it.* Participant 6<br>*With it being all woman and not mixed going to (mainstream drug treatment service provider) and doing groups where men are involved is like, I didn't really want to do it but here because it's all women and I know most of the women that come here, we've all been through it, hence why we all come here. So one way or another we've all been through something that we can all relate to.* Participant 3 |
| Intervention characteristics | *It's [SSW charity] familiar and it's comfortable and it's safe.* Service provider 5<br>*The groups weren't too big, so you sort of—I knew the people that were coming to the groups which was better, so we'd sort of you know built up a rapport.* Participant 4 |
| Impacts of the intervention | *I'm just going to stop [drug use], I'm ready and I'm kind of already preparing for that, so it's kind of brought me to a close, and I mean that as well. Personally it's like, I'm ready, bring it on, I'm like do you know what I've been raped, I've been beaten I've stuck needles in myself(…) I'm done, I'm not playing this game anymore, I deserve better.* Participant 5<br>*It's just made (…) me realise I'm not just a, like, drug addict, sex worker. I'm a real person and I've got feelings and, you know, I've got potential. You know, yeah, they [service providers] build me up a lot.* Participant 6<br>*When she(participant 6)started with [name of intervention] study and she was coming to her Thursday [drug] groups, she didn't want to be associated with street sex working. So she said 'Can you call me (own name rather than working name)?' I could have cried (…). She was owning her own name and taking back ownership of who she is rather than somebody who was street sex working.* Service provider 6<br>*Their chaoticness. (…) To manage that in a [mainstream drug service] group setting would be difficult and I'm not sure how they would manage that. I just know how much regularly how they've turned up [to the intervention drug treatment groups) chaotic and they've turned up leaking out trauma. … I'm far from confident that they would be able to sit under them [mainstream drug service] rules enough to be a part of what it is for here[research study], due to the level of flexibility here and that they would be able to talk about what their problem is without mentioning what they do and that might make them vulnerable* Service provider 7 |

Continued

**Table 2** Continued

| Theme/subtheme | Quotes |
| --- | --- |
| Fidelity | *I guess we were kind of thinking of it in a really linear sense, that the women would engage in the drug groups and then reduce their drug use to then move on to the next group and I'm not sure that that actually happened in reality.*<br>Service provider 5<br>*In the beginning we went in doing the same sort of work that we would do here [mainstream drug services], and it's getting them to look at their behaviour, and the consequences of it and stuff, and it didn't work with these women, it's too much, too direct.* Service provider 10 |

between staff. It was suggested it would have been useful to have collaborative, regular case-review meetings between the services to assess the progress and needs of the participants and enhance the communication channels.

### Participants views on the intervention

Participants described generally positive intervention experiences. They described forming meaningful relationships with the drug group facilitators and clinical psychologist. They liked that the groups were female and sex worker only, they knew other participants already and could speak openly about, and relate to, one another's experience of trauma, drug use and street sex work. Participants also valued that the intervention was delivered at SSW charity premises, which was liked for its familiarity, safety, comfort, convenience and freedom from judgement and shame. The day of the week, time and frequency of sessions, drug group session length and group size were also acceptable to most participants and service providers. These factors overcame some of the barriers participants highlighted to attending mainstream drug services.

### Strengths of the intervention

Through the intervention, participants reflected on their need to address their trauma and drug use. Some acknowledged that they were not ready to address their trauma but aspired to this in future, having had positive experiences of therapy during the intervention. Participants attributed improved well-being, coping strategies and perceptions of self-worth to the intervention. One participant was seen less on the SSW outreach van (an indicator of sex working) and, significantly, stopped using her working name, signifying 'taking back ownership of who she is'. One drug group facilitator felt the flexibility of the study intervention was able accommodate participants' unstable lives and level of trauma that would have prevented them from complying with the rules of conduct in mainstream drug services and thus prevented them from receiving treatment to address their needs. They also felt that an additional positive feature of the intervention was participants being able to discuss their sex work, due to the female SSW only membership of the groups.

Participation in the intervention was described by SSW charity service providers and two participants as supporting and empowering the participants to engage with clinical and support services to address their needs. Another participant felt she used the SSW charity services less now because she needed less support.

### Cost analysis

The total cost of the intervention was £11 710, with staff costs being the largest component (table 4). The most expensive component of the intervention was the 'Getting started' sessions (which totalled £6842). Despite having the second to lowest subtotal cost across the intervention (£1014), the stabilisation groups had the highest cost per session held (£203). Although the one-to-one sessions had the lowest cost per session held (£103), the larger number of sessions at this point resulted in this section having the highest cost per eligible participant (n=4, £724). Trauma screening had the lowest cost per eligible participant (n=5, £191)

### Fidelity

The intervention was broadly delivered as intended incorporating suggested planned changes to the protocol; it was more flexible and less linear than originally planned.[24] Delays in PTSD screening meant that there was only a single drug group, which continued for longer than originally planned.

## DISCUSSION
### Summary of findings

This study used a mixed methods approach to investigate the feasibility and acceptability of a novel, complex intervention to reduce illicit drug use in female drug-dependent SSWs. We demonstrated that drug-dependent SSWs could maintain attendance at female SSW-only drug group sessions and the integrated trauma-focussed treatment approach, in a trusted and supportive environment, with intensive support from SSW charity staff. Recruitment was lower than anticipated, with four participants PTSD screened and whom met criteria for PTSD. They progressed through to the final stage of the intervention; all four participants were ultimately able to access mental health services and one began the process of accessing mainstream drug services. Participants and service providers mostly experienced the recruitment process, the intervention and delivery mechanisms (especially the SSW-only environment) positively. Managing SSW trauma disclosure proved challenging for drug group facilitators

**Table 3** Attendance and retention of participants (top four rows—those that attended trauma screening)

| Open group Getting started | | | | Attended trauma screening | Closed group Getting started | | | | Five mandatory Stabilisation group sessions | | One-to-one sessions | | | | Referred to which services: |
|---|---|---|---|---|---|---|---|---|---|---|---|---|---|---|---|
| Participant | Eligible sessions | Attended sessions* | % Attended | | Eligible sessions | Attended sessions | % Attended | | Attended sessions | % Attended | Treatment | Eligible sessions† | Attended sessions* | % Attended | |
| 1 | 28 | 10 | 36 | Yes | 26 | 4 | 15 | | 3 | 60 | Ext. stabilisation | 8[3] | 1[1] | 13 | Mental health |
| 2 | 27 | 8 | 30 | Yes | 26 | 11 | 42 | | 5 | 100 | EMDR | 12[3] | 4[1]‡ | 33 | Mental health |
| 3 | 20 | 7[2] | 35 | Yes | 26 | 3 | 12 | | 1 | 20 | Ext. stabilisation | 8[4] | 0 | 0 | Mental health |
| 4 | 25 | 19[5] | 76 | Yes | 26 | 15 | 58 | | 4 | 80 | EMDR | 12[2] | 8[4]‡ | 67 | Mental health and mainstream drug |
| 5 | 28 | 5 | 18 | No | | | | | | | | | | | |
| 6 | 28 | 1[1] | 4 | NA | | | | | | | | | | | |
| 7 | 25 | 3 | 12 | NA | | | | | | | | | | | |
| 8 | 23 | 0 | 0 | NA | | | | | | | | | | | |
| 9 | 22 | 0 | 0 | NA | | | | | | | | | | | |
| 10 | 19 | 0 | 0 | NA | | | | | | | | | | | |
| 11 | 19 | 0 | 0 | NA | | | | | | | | | | | |
| Total | | 53 | | 4 | | 33 | | | 13 | | | | 13 | | |

*N session participant arrived late/left early indicated in superscript round brackets.
†N sessions cancelled due to non-attendance in square brackets.
‡Includes one review session.
EMDR, eye movement desensitisation and reprocessing therapy; Ext. stabilisation, extended stabilisation; NA, not actively invited to sessions.

**Table 4** Health economics

| | 1.Getting started | | 2.Trauma screening | | 3.Stabilisation group | | 4.One-to-one sessions | |
|---|---|---|---|---|---|---|---|---|
| Service description | | | | | | | | |
| Session lengths—range | 90–120 min* | | 60 min | | 60 min | | 60–90 min† | |
| Number of sessions held—total | 52 | | 8 | | 5 | | 28 | |
| Eligible participants‡—total | n=11 | | n=5§ | | n=4 | | n=4 | |
| Attendees per session—range | 0–4 | | 1 | | 2–4 | | 1 | |
| Costs | Subtotal £ | £ per ppt | Subtotal £ | £ per ppt | Subtotal £ | £ per ppt | Subtotal £ | £ per ppt |
| A. Staff | | | | | | | | |
| Service manager | £1359.76 | £123.61 | £95.09 | £19.02 | £47.54 | £11.89 | £266.25 | £66.56 |
| Drug group facilitators¶ | £3000.20 | £272.75 | £0.00 | £0.00 | £0.00 | £0.00 | £0.00 | £0.00 |
| Art worker | £123.98 | £11.27 | £0.00 | £0.00 | £0.00 | £0.00 | £0.00 | £0.00 |
| Clinical psychologist | £0.00 | £0.00 | £636.00 | £127.20 | £600.00 | £150.00 | £2040.00 | £510.00 |
| B. Facilities | | | | | | | | |
| Space rental | £1270.50 | £115.50 | £224.00 | £44.80 | £140.00 | £35.00 | £574.00 | £143.50 |
| C. Travel | | | | | | | | |
| Transporting materials | £73.78 | £8.44 | £0.00 | £0.00 | £11.90 | £2.98 | £0.00 | £0.00 |
| Car lifts for service users (petrol) | £38.72 | £3.52 | £2.24 | £0.45 | £3.36 | £0.84 | £1.20 | £0.30 |
| Public transport for participants | £70.20 | £6.38 | £0.00 | £0.00 | £3.90 | £0.98 | £3.90 | £0.98 |
| Taxis for participants | £55.00 | £5.00 | £0.00 | £0.00 | £62.20 | £15.55 | £11.00 | £2.75 |
| D. Materials | | | | | | | | |
| Art supplies | £30.00 | £2.73 | £0.00 | £0.00 | £0.00 | £0.00 | £0.00 | £0.00 |
| Stationary | £0.00 | £0.00 | £0.00 | £0.00 | £5.00 | £1.25 | £0.00 | £0.00 |
| Voucher incentives | £0.00 | £0.00 | £0.00 | £0.00 | £130.00 | £32.50 | £0.00 | £0.00 |
| Refreshments** | £820.00 | £74.55 | £0.00 | £0.00 | £10.00 | £.2.50 | £0.00 | £0.00 |
| Summary | | | | | | | | |
| Total cost | £6842.13 | £622.01 | £957.33 | £191.47 | £1013.90†† | £253.48 | £2896.35 | £724.09 |
| Total cost per session | £131.58 | £11.96 | £119.67 | £23.93 | £202.78 | £50.70 | £103.44 | £25.86 |

| Heat map description: | £0.00 | £0.01–99.99 | | £100–499.99 | | £500.00–1999.99 | | £2000.00 + |
|---|---|---|---|---|---|---|---|---|

*Sessions were originally were 90 min, however when sandwiches were provided drug group facilitators arrived 30 min prior to session to be with participants while they ate.
†One-to-one EMDR sessions were 90 min and one-to-one stabilisation sessions were 60 min.
‡Participants.
§Five participants were eligible for screening; however, only four participants were successfully screened.
¶Getting started groups were facilitated by two drug group facilitators.
**Sandwiches and biscuits.
††Total cost without vouchers would have been £883.90.
EMDR, eye movement desensitisation and reprocessing.

and non-clinical staff and resulted in the recommendation that there is additional training and support for staff in future studies. The need for intervention refinement, for example, extending stabilisation sessions, was suggested to provide additional support prior to trauma treatment. Attendance and adherence barriers primarily related to the issues the intervention sought to address, namely problematic drug use, sex work, and poor mental health, rather than the acceptability of the intervention itself. The total cost of the intervention was £11 710, with staff costs dominating.

### Strengths and limitations of this study
Strengths of our study were that recruiting researchers also conducted the interviews, which may have facilitated a rapport with participants and supported more open and honest reflections. Second, all aspects of the intervention were delivered by specialists which was necessary for this high-risk group with multiple complex comorbidities. Third, PPI formed an important part of the study and informed each stage. Fourth, recruitment over several months with multiple approaches allowed SSWs many opportunities to take part. This approach took account of the rapidly changing lives of SSWs resulting in changing eligibility status as well as allowing time for them to become familiar with the researchers. Finally, recruitment within a trusted agency may have had a positive influence on recruitment.

This feasibility study provides only preliminary information on the intervention performance and cost and does not examine the effectiveness or potential for reducing costs in other parts of the health service or wider society. Delays due to changes to service provision are likely to have adversely influenced study recruitment and retention, with delays resulting in higher service costs, but reflect the real-life issues facing multi-agency work.

## Comparison with other research

This feasibility study is the first, to our knowledge, to address previously highlighted barriers to effective drug treatment for SSWs.[15 22 23] Through incorporating female SSW-only drug groups alongside an intervention with specialised trauma treatment, delivered in a female SSW-only setting by female staff[22 24] we showed how an integrated treatment approach in this complex vulnerable group could feasibly be implemented and delivered, with changes to the intervention, although at a higher than expected cost, mostly due to the delays incurred due to service retendering.

This study is the first interventional study to employ clinical staff from a specialist trauma service to deliver EMDR to address trauma symptoms as part of the drug treatment process for SSWs. These women have been found to have high levels of poor mental health,[16 18] particularly trauma,[5 16 17] which contributes to poor drug treatment outcomes.[7 8] The intervention took account of SSWs frequent experience of abuse and violence,[35 36] and recommendations for female-only trauma-focussed drug treatment interventions[22 37 38] for treatment of PTSD and long-term drug use reduction. Some of the characteristics the intervention sought to address presented as barriers to attendance and retention; however, these are common in studies trying to affect behaviour changes within vulnerable groups.[23 39]

Previous SSW-focussed interventions aiming to reduce drug use[10–14] were unable to demonstrate strong evidence of a positive effect[15] suggesting the need for a novel approach with evidence of efficacy assessed through a robust methodological approach. The highlighted barriers to attendance, engagement and delivery of the intervention are in keeping with other studies[15 23 39] but indicate that further changes to the DUSSK feasibility study design are likely to be required in future studies.

## Conclusion and implications for service provision and research

This study sought to explore the feasibility of delivering a novel complex intervention to a very challenging population with high levels of unmet need. Inclusion criteria were informed by PPI, clinical expertise and academic literature. They targeted women who were likely to benefit the most from a trauma-based intervention but whose drug dependency and chaotic lives made them challenging study participants to recruit and retain. However, all four of the participants screened for PTSD were diagnosed, revealing the unmet need for trauma treatment. Though unsurprising, the severity of trauma disclosed by SSWs proved unexpectedly challenging for service providers.[22 37 38] Further data to understand the extent and severity of PTSD in SSWs are recommended to inform service provision. Overall, the experiences described by those receiving the intervention suggest that it is an acceptable approach to reducing SSWs drug use.

The three services found the intervention valuable and were able to work together effectively despite setbacks such as changing contracts and service pressures. They also suggested more staff support for managing trauma disclosure, extended stabilisation sessions and closer working could improve intervention delivery.

Intervention costs were driven up by poor participant attendance, though staff pressures and the retendering process increased the length (and cost) of the intervention period. However, decreased SSW use of health services, the criminal justice system and impacts of criminal activity on wider society may justify its adoption if future trials demonstrate intervention effectiveness in reducing drug use.

In order to support future interventional trials in this important field where there are few effectiveness studies we recommend the following study refinements for consideration.

1. The intervention could also include women with more stability in their lives to increase recruitment and retention.
2. Regular meetings throughout the study enabling all service providers involved in intervention delivery to express concerns and seek to understand participants needs from the perspective of different professionals so there is effective multiagency support for individual participants where needed.
3. Training for all involved staff in managing the disclosure of trauma.
4. Support and encouragement for participant engagement through provision of transport to and refreshments prior to treatment sessions
5. Intervention flexibility and responsiveness in offering trauma-focussed alternatives to EMDR which may be more suitable for individual participants needs.
6. An extended trauma therapy programme, including extended stabilisation therapy prior to trauma treatment, to accommodate the complexity of SSW needs.

**Author affiliations**
[1]NIHR ARC West, University Hospitals Bristol NHS Foundation Trust, Bristol, UK
[2]Population Health Sciences, Bristol Medical School, University of Bristol, Bristol, UK
[3]Population Health Sciences & NIHR Health Protection Research Unit in Evaluation of Interventions, University of Bristol, Bristol, UK
[4]Population Health Sciences & Health Economics Bristol, University of Bristol, Bristol, UK
[5]Population Health Sciences & Centre for Academic Primary Care, University of Bristol, Bristol, UK
[6]Acer Unit, Blackberry Hill Hospital, Avon and Wiltshire Mental Health Partnership NHS Trust, Bristol, UK
[7]Devon Sexual Health - North Devon, Northern Devon Healthcare NHS Trust, Barnstaple, UK

**Acknowledgements** The authors are extremely grateful to all the women who have participated in the study; all service providers and all other staff whose participation made this study possible. They would like to thank the PPI group for their time, thoughts and suggestions. They are grateful to the following organisations and individuals who have helped with the study for their time, expertise and support: Developing Health and Independence, Bristol Drugs Project, Avon & Wiltshire Mental Health Partnership NHS Trust and One25 charity, Katie Warner, Lucy Pettler, Rosie Davies, Maggie Telfer, Elinor Griffiths, Gill Nowland, Sophie Ramsden, Elaine Driver, Anna Smith, Rhea Warner, Jennifer Riley, Tracey Tudor, Madeline Saunders, Charlotte Hignell, Sophie Banks, Jane Bowman, Sarah Shatwell, Katrina Turner, Hasina Downie, Jo Daniels and Louisa Chowen.

**Contributors** All authors are responsible for the study design, collection of data and analysis. NMR, RP, NJ and JH are responsible for study design, management and coordination. JMK and NJ conducted the interviews. JMK led the qualitative analysis in collaboration with NJ and JH. RP conducted the quantitative analysis. M-JL, RP and JC conducted the costing study. NMR, RP, JMK, M-JL, JC, NJ and JH drafted the paper. All authors read, commented on and approved the final manuscript.

**Funding** The research is supported by a National Institute for Health Research (NIHR) Clinic Trials Fellowship awarded to NJ (NIHR-CTF-2016-05-07), the National Institute for Health Research Collaboration for Leadership in Applied Health Research and Care West (NIHR CLAHRC West P324), now recommissioned as NIHR Applied Research Collaboration West (NIHR ARC West) and Research Capability Funding awarded by University Hospitals Bristol NHS Foundation Trust (RCF 2016-17-18). JMK is partly funded by NIHR Health Protection Research Unit in Evaluation of Interventions.

**Disclaimer** The views expressed are those of the authors and not necessarily those of the NIHR or the Department of Health and Social Care.

**Competing interests** None declared.

**Patient consent for publication** Not required.

**Ethics approval** South West - Frenchay Research Ethics Committee (REC reference: 17/SW/0033; IRAS project ID: 220631). UK HRA approval was on 03/04/2017. The University of Bristol Research Enterprise and Development department sponsored this study (RED reference: RG2756).

**Provenance and peer review** Not commissioned; externally peer reviewed.

**Data availability statement** All data relevant to the study are included in the article or uploaded as supplementary information.

**ORCID iDs**
Rita Patel http://orcid.org/0000-0002-9136-9529
Niamh M Redmond http://orcid.org/0000-0001-8814-3396

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
