## [Reviewer comments · BMJ Open]

ARTICLE DETAILS

TITLE (PROVISIONAL)	Drug Use in Street Sex workers (DUSK) study – results of a mixed methods feasibility study of a complex intervention to reduce illicit drug use in drug dependent female sex workers
AUTHORS	Patel, Rita; Redmond, Niamh; Kesten, Joanna; Linton, Myles-Jay; Horwood, Jeremy; Wilcox, David; Munafo, Jess; Coast, Joanna; Macleod, John; Jeal, Nicola

VERSION 1 – REVIEW

REVIEWER	Dr Alison Munro University of Dundee UK
REVIEW RETURNED	19-Feb-2020

GENERAL COMMENTS	I enjoyed reading this manuscript and found it to be well written, interesting and a well conducted study, so thank you. All sections were well written and structured and the PPI involvement in particular seems to have been a real strength. My own comments revolve around just some points of clarification and general discussion points, stemming from my own interest in recruiting hard to reach groups: Did the authors ever have a target number for recruitment in mind and if so can this be added? 1. In terms of the intervention itself, how long did the intervention sessions last? (In the additional files, Figure 1 seems to be missing which might also give the detail about the length of time of the interventions?2. What was the rationale for involving an art worker and how did that impact do the authors think?3. In terms of the discussion and conclusions, my interest is in what the lessons are around the difficulties of recruitment and retention for future efforts in this important area? Can fewer sessions be offered or shorter sessions? Could additional screening be done to account for 'readiness to change' (behaviour)? Is there any evidence collected regarding the £10 voucher and if increasing that might have boosted recruitment or retention? Was the possibility of peer recruitment ever mentioned in interviews?4. Organisational issues such as service closure and time to appoint staff etc were mentioned so again I'm just interested in what could be done, if anything, to address these sorts of things in future? If it would be possible to say a little bit more about any of these issues around recruiting and retaining this hard to reach group then that would be very valuable. Otherwise, thank you for allowing me to read the paper.
--

REVIEWER	Mohammad Karamouzian University of British Columbia; Vanada
REVIEW RETURNED	01-Mar-2020

GENERAL COMMENTS	This study describes a small-scale intervention aimed at addressing trauma and PTSD among street-based FSW in a UK inner city. While there is merit in interventional studies in this area and I commend the authors for doing the study, their findings, rates of participations, and the way the manuscript is drafted diminished my interest. I hope my suggestions are helpful.  1. The definition of drug dependence is unusual: heroin and/or crack cocaine at least once a week; Why not use a standard definition for substance use disorder? It is unclear why people who use heroin and those who use crack cocaine are combined in the same groups despite the significant differences in substance use profiles of people who are regular heroin users and those who frequently use stimulants such as crack cocaine. 2. How was selling sex defined? What was the timeline for that? Last week (like you did with heroin and crack cocaine) or last year or ever? Or should they have sold sex and used heroin in the past week? 3. The intervention was delivered at SSW charity premises; More details about SSW charity premises; What services do they provide? What are they like? Context? Why did you not try to recruit participants from facilities that provide services to them like DICS or Shelters, etc? 4. The description and write up of the Intervention is very confusing. It seems that parts of the Methods and Results are combined which is unusual in public health scientific writing. 5. Not sure why bullet points are used in the Methods when describing the intervention. This is unusual. 6. "This study is the first interventional study to address trauma symptoms as part of the drug treatment process, identified as important for this group who have been found to have high levels of poor mental health,16 18particularly trauma,5 16 17 which has been highlighted as contributing to poor drug treatment outcomes." I am not sure how novel this idea is as most care and service packages tailored towards street-based FSW include some sort of mental health intervention aimed at PTSD and other mental health conditions. I can think of numerous centres across the globe that I have done research at which provided such services.  7. Sample size is very small and therefore, the analysis lacks enough power or representativeness to make the conclusions worthwhile. 8. "Service providers said working in collaboration with other services was valuable" OR "The unsettled nature of participant's lives was a key attendance barrier." This does not sound like a novel or surprising finding to me.
---

	Accessing care has always been difficult for people who are street-entrenched. Their lifestyle makes it difficult for them to access any intervention. 9. "Participants viewed recruitment as acceptable and experienced the intervention positively" The high proportion of people who declined to participate in the intervention tells a different story about acceptability of the intervention. You do mention some reasons for non-attendance in the study but this deserve further exploration and discussion. 10. "The total cost of the intervention was £11,710, with staff costs dominating" Sounds like a very costly intervention to me for such a small impact. 11. It is unclear what type of mixed-methods approach was utilized here. It is also unclear how the findings of qualitative and quantitative methods were merged to reach the conclusion. 12. The limitations listed are very few. I think this study has numerous limitations that are worth acknowledging. 13. Why have you attached a reporting guideline for a cohort study? This doesn't sound like a cohort observational study to me.
--	--

REVIEWER	Robert Heimer Yale School of Public Health USA
REVIEW RETURNED	28-Mar-2020

GENERAL COMMENTS	There are several areas in which deficiencies in the reporting or in the study design reduce drawing meaningful conclusions from the data collected and analysed. The sample size is small, and the actual number of women who actually engaged with the proposed intervention is smaller yet. The abstract suggests 125 potential participants, but this number is not exactly right. Fourteen women were approached more than once and 84 declined outright, so they cannot be considered potential participants. Full details of eligibility criteria were not provided. Prior to beginning the intervention, participants were required to attend three consecutive group sessions in which facilitators judged when participants were achieving drug use stabilisation. Again, criteria for stabilisation is not specified. This is a very high threshold for continuing on to the next step in the intervention. Despite efforts to coax participation with financial, transportation, and food incentives, intervention could never overcome intermittent or terminated participation. This was blamed on "participant's [sic] unstable behavior." They did not test for or consider the possibility that low rate of continued participation was the lack of fit between the intervention and the participants' needs. It is noteworthy that the clinical psychologist cited in the manuscript "...suggested that without the 're-traumatising' effects of street sex work, the effectiveness of the trauma processing in the trauma treatment might be enhanced." Since the intervention does nothing to reduce the street sex work burden on the target audience for the intervention, it is hard to envision how the intervention might prove effective even if were more acceptable and participation rates were
--

	increased greatly. There is a section entitled “Impacts of the intervention” that rather than describing impacts lists reasons why participants felt that the intervention was likely to have very limited impact. The conclusions do not follow on the results. It is perplexing that the study design was intended to address those with multiple problems especially longstanding trauma and chaotic living conditions, but the conclusions suggest modifying the target audience towards “includ[ing] women with more stability in their lives to increase recruitment and retention.” In doing so, they indicate the unsuitability of their intervention for those they seek to assist.
--	---

REVIEWER	Dr Amanda Roxburgh Burnet Institute Australia
REVIEW RETURNED	25-May-2020

GENERAL COMMENTS	This is an interesting piece of work that tackles an incredibly important issue – addressing drug use and PTSD among street-based sex workers. I think the authors need to reframe the paper to better reflect their results. I make comments by section below: Abstract Objectives Can the authors please specify a little more clearly that the aims of the intervention were targeting both drug use and PTSD. This wasn't clear from the abstract. Results I think the authors need to state clearly in the abstract how many of the women completed the entire intervention. There are two conflicting statements in the manuscript in relation to this: Page 11 under group attendance line 31 and 32: “All four participants missed at least two consecutive trauma treatment appointments and were deemed to have withdrawn from the sessions” Page 14 under Discussion – line 35: “They progressed through all stages of the intervention and all four participants were ultimately able to access mental health services . . .” The first statement suggests none of the participants completed the entire intervention and the second statement suggests all four completed the intervention. Conclusions The conclusions in the abstract need to align more with the findings – the difficulties experienced in both recruitment and retention, and the substantial costs involved make this model of treatment less viable. Introduction As per above comment, page 4 lines 46 and 47 – can the authors be more explicit here that the intervention is targeting drug use and
---

	trauma treatment. Methods PPI Can the authors describe how this co-design operated. How did they invite peers to be involved? How did they contribute to study design etc. A little more detail on the process of peer involvement. The intervention Page 5 - line 52 Sentence starting “The intervention consisted of SSW only drug treatment groups” Should this read “SSW drug treatment only groups”? Can the authors provide more details about what drug treatment groups involved? Page 5 – line 53 “all delivered by female staff” – Should this read female psychologists? Page 6 – line 7 “when group facilitators judged participants were achieving drug use stabilisation.” Can the authors describe what criteria were used to determine ‘stable drug use’? Page 6, line 9 – Details about the PTSD checklist. Can the authors move this up to the start of the intervention section, after they detail the drug treatment and before they mention screening for PTSD. Also, further detail of the score used to determine whether participants screened positive for a provisional diagnosis of PTSD would be good. Terminology in relation to PTSD is also important – can the authors ensure that this is consistent throughout the manuscript? Are they talking about the experience of PTSD symptoms or a provisional PTSD diagnosis? Data analysis Statistical analysis Can the authors describe here what software they used for descriptive statistics? Was it SPSS, excel, something else? Cost analysis Were staff costs factored in even when sessions didn’t go ahead due to non-attendance? Was this an expense for the study? If so, this should be articulated. Results Were any other demographic details collected on participants? Education? Housing etc? Recruitment and attendance Suggest this sub-heading be renamed Recruitment and retention to better align with the aim of assessing the feasibility of recruitment and retention of participants. Group attendance Page 11 lines 31 and 32 “All four participants missed at least two consecutive trauma treatment appointments and were deemed to
--	--

have withdrawn from the sessions”

As per my above comment does this mean no participants completed the intervention? If so, can the authors articulate this more clearly.

Discussion

Summary of findings

Page 14 line 35 Sentence “They progressed through all stages of the intervention and all four participants were ultimately able to access mental health services . . .”

As per above comments, this seems to contradict the statement the none of the participants moved through all stages of the intervention. Can the authors please clarify?

Comparison with other research

Page 15 lines 30 & 31 “ . . we showed how an integrated treatment approach in this complex vulnerable group can be feasibly implemented and delivered.”

I’m not sure the authors can make this statement as their results suggest that the treatment wasn’t feasibly implemented given low numbers (perhaps none) going through to completion. I suggest this sentence be amended to more accurately reflect their findings.

Conclusions and implications

Again, I think the authors need to make comment on the need for a different approach to addressing drug use and PTSD among SSW as the interventions they investigated, while seemingly acceptable to the women, were not very well attended.

Perhaps leading on from this, given their findings could the authors reflect on how services might better support SSW in relation to their drug use and PTSD in more informal ways?

Table 1

Can the authors add a line in this table for Screened positive for PTSD or Experiencing PTSD symptoms (whichever terminology they are using throughout the manuscript)?

Table 3

This table would be clearer if the authors highlighted only those women who attended trauma screening, in order to differentiate the 2 groups.

Overall comment

I’m not sure if I’ve misunderstood the findings but from my reading it appears that the approach the authors investigated was not a feasible one to address drug use and PTSD among SSW. This is not to say that the work is not valuable. Even showing that the approach was not feasible is an important advance in understanding how to target drug use and PTSD among this group. It may lead to more helpful ways that services might respond to these issues in a more sustainable way (outside of research funding and within service

	resources and capacity). There are also likely benefits that the women attained from engaging in this program. All of these issues could be incorporated into the discussion to provide different ways forward. Finally, did the authors include any feedback mechanism of the study to the women in relation to their findings? If so could they include this in the paper. CONSORT CHECKLIST Generalisability – this is not addressed within the manuscript. I have commented in my review above on gaps in other rep
--	--

VERSION 1 – AUTHOR RESPONSE

Reviewer 1 comments	Responses
I enjoyed reading this manuscript and found it to be well written, interesting and a well conducted study, so thank you. All sections were well written and structured and the PPI involvement in particular seems to have been a real strength. My own comments revolve around just some points of clarification and general discussion points, stemming from my own interest in recruiting hard to reach groups: 1(a) Did the authors ever have a target number for recruitment in mind and if so can this be added?	We thank the reviewer for their positive comments. Yes, we had an initial target of 30 participants who we expected might start the intervention with an attrition of at least a third of those. This was published in the protocol paper (https://pubmed.ncbi.nlm.nih.gov/30391916/) which we referenced at the end of the Introduction (page 4 paragraph 4). We have made this clearer in the new version at the start of the Methods section (page 5) and we refer specifically to the sample size on page 5, second paragraph (track changes version).
1. In terms of the intervention itself, how long did the intervention sessions last? (In the additional files, Figure 1 seems to be missing which might also give the detail about the length of time of the interventions?)	The intervention time sessions are listed in table 4. We checked in the journal's author centre and Figure 1 (participant flow chart) was included in the original submission, however there appeared to be a page gap, which may explain why it was missing. We have re-submitted this figure.
2. What was the rationale for involving an art worker and how did that impact do the authors think?	Due to a process of service retendering and change of drug service provider (which was not related to the study and was out of our control), along with staffing issues within the service providers, the 'getting started' sessions continued beyond the schedule originally planned. The group facilitators sought additional materials for the sessions to maintain interest and usefulness for participants, which, as DUSK was a feasibility study, was an opportunity to refine/tweak the intervention. The drug group facilitators had found an art therapist helpful in

	other groups to enable expression of ideas/feelings that may otherwise struggle to vocalise and therefore invited the art therapist to 'getting started' sessions. Participants received this positively anecdotally and via the interviews. Page 6, paragraph 2 (bullet points) has been modified to explain this more clearly.
3. In terms of the discussion and conclusions, my interest is in (i) what the lessons are around the difficulties of recruitment and retention for future efforts in this important area? (ii) Can fewer sessions be offered or shorter sessions? (iii) Could additional screening be done to account for 'readiness to change' (behaviour)? (iv) Is there any evidence collected regarding the £10 voucher and if increasing that might have boosted recruitment or retention?	(i) The problems around recruitment and retention predominantly related to the women's lives and these issues could be addressed by extending inclusion criteria to include women with greater life stability. (ii) We acknowledge that there were more 'getting started' sessions than originally planned due to the retendering process, service provider change and staffing delays so yes fewer sessions could have been offered. Our interviews did highlight that some participants may have appreciated shorter sessions. Regarding the trauma sessions, this is unlikely; the clinical psychologist suggested increasing the number of sessions beyond the usual number delivered by the NHS due to the complexity of addressing trauma in this group. They had already reduced the session duration from two hours to 90 minutes. (iii) Possibly – screening, as is typical in studies and trials, was already a balance between data burden to the participant before informed consent and assessing eligibility to the study. We had discussed the screening aspects with the PPI group. We attempted to derive objective criteria for drug group participants, which accurately reflected a 'readiness to change', but found individual assessment was required, drawing on the skills and experience of the drugs workers. The sex worker charity staff did support promotion of the study to women whom they felt may benefit but we were keen to ensure that minimal bias in terms of invitations were maintained, so we approached as many women during recruitment sessions as possible. In addition, the inclusion criteria did encourage sex workers who may have had the most chaotic lives. We agree that a 'readiness to engage and change behaviour' may be a good criterion to assess and this will be looked at further if a future trial is funded. (iv) We had agreed £10 vouchers with the intervention delivery partners plus we were restricted to this by the local research Ethics

(v) Was the possibility of peer recruitment ever mentioned in interviews?	committee. These were used as a research tool to encourage attendance and allow opportunity for the intervention feasibility to be explored, were appreciated by participants, and staff felt they motivated attendance. Use of financial incentives has been demonstrated in other studies.² These are unlikely to be given to patients in mainstream services if the intervention were to be rolled out as routine care. We had previously given £20 vouchers to SSW research participants for in-depth interviews but that, unlike the intervention, has no direct benefit to recruiting participants. (v) Yes we did discuss this with the PPI group but they were against peer recruitment.
4. Organisational issues such as service closure and time to appoint staff etc were mentioned so again I'm just interested in what could be done, if anything, to address these sorts of things in future? If it would be possible to say a little bit more about any of these issues around recruiting and retaining this hard to reach group then that would be very valuable. Otherwise, thank you for allowing me to read the paper.	This is a very important point and we thank the reviewer for highlighting this. Unfortunately, these issues represent the reality for mental health services, which are chronically under-funded. Constant service retendering has a negative impact on service planning, continuity of delivery and staffing. Funding for the intervention would need to be ring-fenced to protect the intervention and currently that seems unlikely to happen. We are constrained by word limits in this paper, however we are drafting another paper which will focus on issues affecting intervention delivery. We are grateful to this reviewer for their insightful and interesting comments.

Reviewer 2

Reviewer 2 comments	Responses
	Please note: We have responded to this reviewer's comments however our appeal letter (submitted on 20th April 2020) focussed on our concerns with this reviewer's comments.
This study describes a small-scale intervention aimed at addressing trauma and PTSD among street-based FSW in a UK inner city.	 • DUSK is not a small scale intervention study, it is a mixed methods (quantitative and qualitative) feasibility study (precursor to a trial) as described in the published protocol paper (Jeal N, Patel R, Redmond NM, et al. Drug use in street sex workers (DUSK) study protocol: a feasibility and acceptability study of a complex intervention to reduce illicit drug use in drug-dependent female street sex workers. BMJ Open 2018;8(11):e022728; https://pubmed.ncbi.nlm.nih.gov/30391916/) We have made reference to this paper at the

While there is merit in interventional studies in this area and I commend the authors for doing the study, their findings, rates of participations, and the way the manuscript is drafted diminished my interest. I hope my suggestions are helpful.	start of the Methods on page 5 (track changes version), first paragraph to make this clearer (this was originally referred to at the end of the introduction on page 4).  • The findings and rates of participation are consistent with other mixed methods feasibility studies. • The manuscript was drafted in line with the author guidelines while also adhering to the word count. We attempted to reduce the methods content due to word count limitations, as the methods are described in detail in the study's published protocol paper. As described above, we have made reference to this in a clearer way on page 5.
1. (i) The definition of drug dependence is unusual: heroin and/or crack cocaine at least once a week; Why not use a standard definition for substance use disorder? (ii) It is unclear why people who use heroin and those who use crack cocaine are combined in the same groups despite the significant differences in substance use profiles of people who are regular heroin users and those who frequently use stimulants such as crack cocaine.	(i) This definition was chosen as this is the commonly used type of classification in sex work research (Jeal N, Macleod J, Turner K, Salisbury C Systematic review of interventions to reduce illicit drug use in female drug-dependent street sex workers. BMJ Open 2015;5:e009238.doi:10.1136/bmjopen-2015-009238) and the data collection forms were based on standard core data sets published by the UK Dept. of Health's Public Health England in 2016 entitled "July consultation on proposed amendments to the data set collected on alcohol and drug treatment by the National Drug Treatment Monitoring System (NDTMS)" a shortened URL to this document is available here https://tinyurl.com/y9wuhxrd. Finally, if the study were to go forward to a RCT, this definition would facilitate comparison of the intervention's efficacy, outcomes and how it could be applied in practice. They are combined because UK SSWs tend to use both drugs and this is displayed clearly already in Table 1. Although the following link is a recent publication, it has been well established in the Bristol region that people who inject drugs use both heroin and crack cocaine: (http://www.bristol.ac.uk/policybristol/policy-briefings/bristol-in-brief-1-drugs-in-the-south-west/)
2. (i) How was selling sex defined? (ii) What was the timeline for that? Last	(i) The exchange of money for sexual services is an internationally recognised/accepted definition so does

week (like you did with heroin and crack cocaine) or last year or ever? Or should they have sold sex and used heroin in the past week?	not really need formally defining within the paper. (ii) Table 1 shows this information. We have also clarified this further on page 5, first paragraph.
3. The intervention was delivered at SSW charity premises; (i) More details about SSW charity premises; What services do they provide? What are they like? Context? (ii) Why did you not try to recruit participants from facilities that provide services to them like DICs or Shelters, etc?	(i) Due to word count limit, we are only able to add that the SSW charity premises supplies support, health and advocacy services. Further details are in the protocol paper (as referenced). (ii) The first line of the recruitment section (page 5, second paragraph) says Local organisations that SSWs were known to access were provided with study promotional materials. The SSW charity was chosen as the location as it was a safe place for women to attend, free from fear of or intimidation from other drug users, sex work clients etc. whom are known to prevent and specifically hinder sex workers from engaging with drug addiction/prevention services. The protocol paper details that shelters and homeless organisations were offered and given flyers to recruit women. As mentioned above, we have made reference to the protocol paper at the start of the methods (page 5 first paragraph).
4. The description and write up of the Intervention is very confusing. It seems that parts of the Methods and Results are combined which is unusual in public health scientific writing.	We have referenced the protocol paper early in the Methods section on page 5, as described above, which we believe will draw reader's attention to this in a clearer way. Originally on page 6 (end of first paragraph), we had stated that the intervention proceeded as described but to increase clarity we have separated out this last sentence into a new paragraph and clarified it was the published protocol that was followed. As this was a mixed methods feasibility study the methods section includes a bulleted list of changes that took place during the study implementation phase so changes are clear to readers. We have removed anything that may look like results from the method section.
5. Not sure why bullet points are used in the Methods when describing the intervention. This is unusual.	The section with bullet points was to highlight the changes to the methods described in the protocol paper, so that it was clear what was performed, working within the journal word limit.
6. "This study is the first interventional study to address trauma symptoms as part of the drug	EMDR delivered by trauma specialists working in a specialist trauma service has not previously

treatment process, identified as important for this group who have been found to have high levels of poor mental health,16 18particularly trauma,5 16 17 which has been highlighted as contributing to poor drug treatment outcomes.” I am not sure how novel this idea is as most care and service packages tailored towards street-based FSW include some sort of mental health intervention aimed at PTSD and other mental health conditions. I can think of numerous centres across the globe that I have done research at which provided such services.	been used to treat PTSD in female street-based street sex workers as part of a drug treatment package. Services frequently offer mental health support but ‘mental health intervention’ varies greatly in content and training/qualifications of delivery staff. The novel aspects here are:  - specialist staff from a specialist clinical trauma service - use of EMDR as a treatment modality We have modified paragraph 2 on page 15 (Comparison with other research section) to better reflect this. We have searched research databases and there are no recent research studies (observational, retrospective or trials), to our knowledge that include specialist clinical staff employing EMDR to treat PTSD as a means to reducing illicit drug use. It would have been very helpful if the reviewer had referenced the centres or studies that he is familiar with, as we would be very willing to review these published studies and include them in this paper as references.
7. Sample size is very small and therefore, the analysis lacks enough power or representativeness to make the conclusions worthwhile.	Again as previously pointed out, this is because it is a feasibility study and this is normal, routine and accepted in feasibility studies. The intention of feasibility studies is not to have enough statistical power to show effectiveness, but to understand processes like recruitment and acceptability of the intervention, for example.
8. “Service providers said working in collaboration with other services was valuable” OR “The unsettled nature of participant’s lives was a key attendance barrier.” This does not sound like a novel or surprising finding to me. Accessing care has always been difficult for people who are street-entrenched. Their lifestyle makes it difficult for them to access any intervention.	The intervention involved statutory and non-statutory services from clinical and non-clinical backgrounds, which traditionally would find it difficult to work so closely together because of different organisational cultures, different funding streams and patient confidentiality issues. Although providers aspire to work well across organisations, it is not always a given that this will be executed effectively.³ Therefore, for the services it was a novel way of working which proved valuable. This may be unique to the UK setting. The intervention was developed based on a systematic review, qualitative work with SSWs and service providers as well as consultation with experts in the field. It was hoped that some of the novel aspects of the intervention, such as SSW-only groups and non-judgemental/supportive

	setting, would remove some of the traditional barriers cited by participants as preventing access to mainstream drug services. Therefore, it was disappointing and a significant finding for the researchers that participants lifestyle still represented a barrier to access.
9. "Participants viewed recruitment as acceptable and experienced the intervention positively" The high proportion of people who declined to participate in the intervention tells a different story about acceptability of the intervention. You do mention some reasons for non-attendance in the study but this deserve further exploration and discussion.	This is a conclusive point from the qualitative aspect of the study, where those who did participate found the recruitment process acceptable. The high decline numbers are from the quantitative aspect and potentially represent numerous approaches to the same women, by different recruiters, as indicated in Figure 1 "women spoken to at recruiting site". However we accept that it is not clear this meant the same women were approached multiple times over the recruitment period. We have made changes to the methods on page 5 paragraph 2, page 10 paragraph 2 and Figure 1. Our qualitative work did not explore reasons for non-participation among those who declined recruitment. It may have been advantageous to have interviewed women about the reasons for declining and this is something we will build into a future, larger trial.
10. "The total cost of the intervention was £11,710, with staff costs dominating" Sounds like a very costly intervention to me for such a small impact.	It was a feasibility study and efficacy was not assessed. Our cost analysis indicates that 'staff' represented the largest cost within the intervention. Assessing the impact of the intervention was out of scope. The economic analysis focuses on costs (without comparison to outcomes), to solely profile how costs differ across the different sections of the intervention.
11. It is unclear what type of mixed-methods approach was utilized here. It is also unclear how the findings of qualitative and quantitative methods were merged to reach the conclusion.	The mixed methods approach was described in the protocol paper, as previously stated. We have added to page 8, paragraph 3, of the methods section "The integration of qualitative and quantitative data used the established 'following a thread' technique where key themes were traced using all data sets.
12. The limitations listed are very few. I think this study has numerous limitations that are worth acknowledging.	We feel this is a moot point because it appears the design of the study has been misunderstood and we have defended criticisms in relation to the design above. Due to the word limit we have not been able to extend this section in the paper.
13. Why have you attached a reporting guideline	BMJ Open author guidelines request completed

for a cohort study? This doesn't sound like a cohort observational study to me.	standard study checklist protocols and, as there is not a specific one for feasibility studies, we chose this one as it is the closest one to a feasibility study. We provided this for transparency and research standards purposes.
---	---

Reviewer 3

Reviewer 3 comments	Responses
	Please note: We have responded to this reviewer's comments however our appeal letter (submitted on 20 th April 2020) focussed on our concerns with this reviewer's comments.
There are several areas in which deficiencies in the reporting or in the study design reduce drawing meaningful conclusions from the data collected and analysed. 1. The sample size is small, and the actual number of women who actually engaged with the proposed intervention is smaller yet.	The sample size is small because this is a feasibility study designed to assess feasibility and acceptability of the intervention and study design. This is normal, routine and accepted in feasibility studies. The intention of feasibility studies is not to have enough statistical power to show effectiveness, but to understand processes like recruitment and acceptability of the intervention, for example. The feasibility study design and protocol has previously been peer reviewed and published, is referred to in the introduction and was followed during the study execution (Jeal N, Patel R, Redmond NM, et al. Drug use in street sex workers (DUSK) study protocol: a feasibility and acceptability study of a complex intervention to reduce illicit drug use in drug-dependent female street sex workers. BMJ Open 2018;8(11):e022728; https://pubmed.ncbi.nlm.nih.gov/30391916/).
2. The abstract suggests 125 potential participants, but this number is not exactly right. Fourteen women were approached more than once and 84 declined outright, so they cannot be considered potential participants.	This is a valid point in that potential participants could be misconstrued. We have made changes to the results section of the abstract (page 2), the results section in the main text (page 10 paragraph 2) and Figure 1 to better reflect this.
3. Full details of eligibility criteria were not provided.	The full eligibility criteria are in the method sections on page 5 first paragraph. These have been subsequently modified in response to reviewer 2's second numbered point.
4. (i) Prior to beginning the intervention, participants were required to attend three consecutive group sessions in which facilitators judged when participants were achieving drug use stabilisation.	(i) The drug group sessions were part of the intervention. (ii) The protocol paper provides more details about the methods, and this is referred to in the introduction and now, more clearly at the

(ii) Again, criteria for stabilisation is not specified. (iii) This is a very high threshold for continuing on to the next step in the intervention.	start of the methods due to word count restrictions (page 5 first paragraph). The protocol paper states that “After attending four group meetings, if participants are perceived by the group facilitators as exhibiting evidence of life/drug use stability such as engagement and functioning in the group, positive interaction with group facilitators, regular OST, they will be offered transfer to the ‘Preparation for Recovery’ group.” As stated in the methods in this paper, on page 6, participants attended three consecutive sessions and the bullet points detail the changes made from the protocol paper and why. We have made minor changes to the bullet points on page 6 for further clarity (additional changes have been made in response to reviewer 1’s comments). (iii) Yes this was a high threshold, and an opportunity to see if this was possible within a feasibility study, instead of a full effectiveness trial. Stability was required to enable participants to engage with trauma therapy (which mainstream drug services typically require) which can be quite challenging. Drug group facilitators also liaised with the clinical psychologist in assessing readiness for trauma treatment.
5. Despite efforts to coax participation with financial, transportation, and food incentives, intervention could never overcome intermittent or terminated participation. This was blamed on “participant’s [sic] unstable behavior.” They did not test for or consider the possibility that low rate of continued participation was the lack of fit between the intervention and the participants’ needs.	Participants were not coaxed with financial/transport or food incentives – these were approved by the ethics committee to encourage women to attend, because women did not have transport or cash to get themselves to the study group sessions, even though they provided informed consent to participate. As this was a feasibility study, recruitment and attendance to the intervention were stated in our aims – we wanted to understand whether recruiting and the retention of women to the study was feasible. It is well described in sex worker literature how SSWs’ lives can be chaotic with unstable living conditions and transport arrangements. We disagree with the reviewer’s cynical opinion that they were coaxed and blamed – conducting research with vulnerable women such as sex workers requires tailored approaches to reflect their lives and encourage participation – SSWs are an under-researched group, probably because of these and other challenges. The reviewer appears to have little or no knowledge of the existing sex worker literature.

	The intervention was developed over a long period of time, drawing on literature⁴ as well as qualitative work with sex workers^{5,6} and working with experts in the field (SSW charity, NHS trauma service, drug services, clinicians) who have longstanding hands-on experience of working with and providing care for this group. Research has tended to exclude the most chaotic groups because they are hard to engage and follow up. This is all stated in the introduction within the limited word count. There was lively debate within the research team and delivery partners about the feasibility of engaging with the most chaotic SSWs. It was decided that the study had to explore the feasibility of using the intervention with the most street entrenched because they tend to be excluded from studies and stood to have the greatest health gain if the intervention was successful. In light of our learning from the feasibility study, a future trial is likely to extend the criteria to include women who are slightly further along the journey to recovery, likely still within 3-6 months of street entrenchment, who are still at significant risk of slipping back into street sex work which tends to run a relapsing and remitting course.⁷
6. It is noteworthy that the clinical psychologist cited in the manuscript "...suggested that without the 're-traumatising' effects of street sex work, the effectiveness of the trauma processing in the trauma treatment might be enhanced." Since the intervention does nothing to reduce the street sex work burden on the target audience for the intervention, it is hard to envision how the intervention might prove effective even if were more acceptable and participation rates were increased greatly.	As previously stated this was a feasibility study and its aims were not to test the effectiveness of the intervention, but rather to test the processes of whether recruitment and retention to the intervention were feasible. We do not know if the intervention would prove effective at this present time, that would be best assessed in an effectiveness trial – this study does however show us that we can recruit some sex working women to these female only groups and that they can move through the intervention successfully – so an effectiveness trial is, in part, a possibility. The women-only groups were a novel element of the intervention, which is usually prohibited in mainstream services due to mixed groups. The lives of street sex workers can be viewed as a 'work-score-use' cycle;⁵ working for money to buy drugs, then using drugs and back out to working again. The driver for the cycle is the drug use rather than the sex work. By reducing drug use involvement in sex work will reduce. Trying to do it the other way round is impossible.

7. There is a section entitled “Impacts of the intervention” that rather than describing impacts lists reasons why participants felt that the intervention was likely to have very limited impact.	We have considered this comment carefully and reflected that these are more in line with strengths of the intervention. We have renamed the section ‘Strengths of the intervention ‘ and restructured one sentence, which, on re-reading, was not clearly conveying the message. The changes made are on page 13, paragraph 2.
8. The conclusions do not follow on the results.	Overall we believe the conclusions follow from the results and are in line with a mixed methods feasibility study. However we have made changes in the conclusions for clarity purposes (page 15-16).
9. It is perplexing that the study design was intended to address those with multiple problems especially longstanding trauma and chaotic living conditions, but the conclusions suggest modifying the target audience towards “includ[ing] women with more stability in their lives to increase recruitment and retention.” In doing so, they indicate the unsuitability of their intervention for those they seek to assist.	The aim of the intervention was initially to address those with the most chaotic lives as it was deemed they might benefit most if the intervention were ultimately effective (in a future trial). The feasibility study was aiming to explore whether, against the odds, those women could be (a) recruited and (b) if they could engage with the intervention We acknowledge the tension of chaotic women versus more stable women. The quote provided by the reviewer is one option for a future effectiveness study.

Review 4:

Reviewer 4 comments	Responses
This is an interesting piece of work that tackles an incredibly important issue – addressing drug use and PTSD among street-based sex workers. I think the authors need to reframe the paper to better reflect their results. I make comments by section below: Abstract Objectives Can the authors please specify a little more clearly that the aims of the intervention were targeting both drug use and PTSD. This wasn't clear from the abstract. Results I think the authors need to state clearly in the abstract how many of the women completed the entire intervention. There are two conflicting statements in the manuscript in relation to this: Page 11 under group attendance line 31 and 32: “All four participants missed at least two	We thank the reviewer for their comments. This was a feasibility study to see if an intervention to address drug use and PTSD among street-based sex workers was possible. The aims were to understand if recruitment and retention of participants was possible, to examine the intervention experiences and acceptability and to explore the costs. We have modified the abstract to clarify the relationship between drug use and trauma symptoms and therefore the rationale of the intervention approach. We can see how this can be read as conflicting. All four women did reach the last stage of the intervention and attended one-to-one sessions but did not fully complete all sessions. We have made changes to the abstract to clarify the

consecutive trauma treatment appointments and were deemed to have withdrawn from the sessions” Page 14 under Discussion – line 35: “They progressed through all stages of the intervention and all four participants were ultimately able to access mental health services . . . “ The first statement suggests none of the participants completed the entire intervention and the second statement suggests all four completed the intervention. Conclusions The conclusions in the abstract need to align more with the findings – the difficulties experienced in both recruitment and retention, and the substantial costs involved make this model of treatment less viable. Introduction As per above comment, page 4 lines 46 and 47 – can the authors be more explicit here that the intervention is targeting drug use and trauma treatment. Methods PPI Can the authors describe how this co-design operated. How did they invite peers to be involved? How did they contribute to study design etc. A little more detail on the process of peer involvement. The intervention Page 5 - line 52 Sentence starting “The intervention consisted of SSW only drug treatment groups” Should this read “SSW drug treatment only groups”? Can the authors provide more details about what drug treatment groups involved?	numbers reaching the final intervention stage (within the word limit). On page 11, paragraph 2 (track changes version) we have made changes to clarify their attendance and then withdrawal (in line with the protocol) On page 14 paragraph 2, we have made small changes to this sentence to be consistent with the message. We feel the conclusions do reflect this and the nature of a feasibility study, and are in line with the results. Whilst the cost of the intervention may appear substantial, if, in future, it proved effective the cost savings through SSWs reduced unscheduled use of health services, the criminal justice system and impacts of acquisitive criminal activity on wider society may justify its rolling out. We are constrained by word limit and have made minor changes to the conclusions. We have made changes to page 4 last paragraph, to make it clear the intervention is related to trauma. We are mindful that the intervention is targeting drug use and trauma symptoms but that the study was to see if the intervention processes were feasible and not an effectiveness study. The PPI group was initially convened with the help of the sex worker charity. The PPI process is explained in the referenced published protocol paper (https://pubmed.ncbi.nlm.nih.gov/30391916/) which was referred to in the introduction and has been referred to more clearly at the beginning of the Methods section now (page 5, first paragraph). “Intervention and study design was developed based on input from the study PPI group that included women currently and previously involved in SSW and illicit drug use. The group convened before and during set up, contributed to the protocol development as well as the design of participant facing study documentation. Subsequent meetings have informed recruitment, topic guides, plain language study summary and plans for study
--	---

Page 5 – line 53 “all delivered by female staff” – Should this read female psychologists? Page 6 – line 7 “when group facilitators judged participants were achieving drug use stabilisation.” Can the authors describe what criteria were used to determine ‘stable drug use’? Page 6, line 9 – Details about the PTSD checklist. Can the authors move this up to the start of the intervention section, after they detail the drug treatment and before they mention screening for PTSD. Also, further detail of the score used to determine whether participants screened positive for a provisional diagnosis of PTSD would be good. Terminology in relation to PTSD is also important – can the authors ensure that this is consistent throughout the manuscript? Are they talking about the experience of PTSD symptoms or a provisional PTSD diagnosis? Data analysis Statistical analysis Can the authors describe here what software they used for descriptive statistics? Was it SPSS, excel, something else?	dissemination. Ongoing PPI meetings will focus on troubleshooting issues identified during the study process and at the end of the study will focus on interpretation of results and dissemination methods.” We have kept the terminology consistent with the published protocol paper. Further details are available in the protocol paper, which is now highlighted more clearly at the start of the Methods section, page 5, first paragraph. This was originally referred to in the introduction (final paragraph) but we recognise that this could be missed by the reader. No, the drug groups were not delivered by psychologists but by specialist drug facilitators trained in preparing, hosting and running drug addiction group sessions (from the drug treatment charity which was successful in securing the contract for city-wide community drug service provision during the re-tendering process). The protocol paper provides more details about the methods as we could not recount all details of the methods fully due to word limitations. The protocol paper states that “After attending four group meetings, if participants are perceived by the group facilitators as exhibiting evidence of life/drug use stability such as engagement and functioning in the group, positive interaction with group facilitators, regular OST, they will be offered transfer to the ‘Preparation for Recovery’ group.” This has been done as requested (page 5 last paragraph). Again this is detailed in the protocol paper and is not included here due to word limitations. “Participants will be individually screened for currently experiencing PTSD symptoms in a 90 min one-to-one session with a registered
---	---

Cost analysis Were staff costs factored in even when sessions didn't go ahead due to non-attendance? Was this an expense for the study? If so, this should be articulated. Results Were any other demographic details collected on participants? Education? Housing etc? Recruitment and attendance Suggest this sub-heading be renamed Recruitment and retention to better align with the aim of assessing the feasibility of recruitment and retention of participants. Group attendance Page 11 lines 31 and 32 "All four participants missed at least two consecutive trauma treatment appointments and were deemed to have withdrawn from the sessions" As per my above comment does this mean no participants completed the intervention? If so, can the authors articulate this more clearly. Discussion Summary of findings Page 14 line 35 Sentence "They progressed through all stages of the intervention and all four participants were ultimately able to access mental health services . . ." As per above comments, this seems to contradict the statement the none of the participants moved through all stages of the intervention. Can the authors please clarify? Comparison with other research Page 15 lines 30 & 31 " . . we showed how an integrated treatment approach in this complex vulnerable group can be feasibly implemented and delivered." I'm not sure the authors can make this statement as their results suggest that the treatment wasn't feasibly implemented given low numbers (perhaps none) going through to completion. I suggest this sentence be amended to more accurately reflect their findings.	female clinical psychologist. The session will consist of a clinical interview to elicit information about symptoms related to the diagnostic criteria for PTSD as stated in the American Psychiatric Association's Diagnostic and Statistical Manual of Mental Disorders 5 (DSM-5).³⁸ The PTSD Check List-5 (PCL-5) is used to assist the clinical assessment, provide a baseline score and provide the clinical psychologist with a provisional PTSD diagnosis. If the participant is found to be currently experiencing PTSD symptoms, she will be offered a place in the 'Stabilisation' group. If she is deemed to not benefit from the 'Stabilisation' group, she can continue in the 'Preparation for recovery' group with eventual referral to mainstream drug services." Throughout the intervention and until assessment by the psychologist we are referring to the experience of PTSD symptoms. We have checked through the manuscript to ensure consistency as requested and made appropriate changes. Stata 14 was used for this. We have added this (page 8, paragraph 4).  - Getting started: Staff waited for a maximum of 45 minutes to see if any attendees arrived. If no attendees arrived the staff left, to undertake other work and were thus only costed for the first section of the session. - Trauma screening and 1-1s: Staff costs were factored in even when sessions didn't go ahead (non-attendance) due to a contractual agreement with the psychologist . - Stabilisation group: No scheduled stabilisation groups had non-attendances. No, we collected only the data we required to answer the aims of the feasibility study and additional demographics beyond age, sex and permitted contact details were not required. If the intervention were to go to a full effectiveness trial, then collection of further demographic information may be ethical and warranted. We were mindful of the data burden to participants.
--	---

Conclusions and implications Again, I think the authors need to make comment on the need for a different approach to addressing drug use and PTSD among SSW as the interventions they investigated, while seemingly acceptable to the women, were not very well attended. Perhaps leading on from this, given their findings could the authors reflect on how services might better support SSW in relation to their drug use and PTSD in more informal ways? Table 1 Can the authors add a line in this table for Screened positive for PTSD or Experiencing PTSD symptoms (whichever terminology they are using throughout the manuscript)? Table 3 This table would be clearer if the authors highlighted only those women who attended trauma screening, in order to differentiate the 2 groups. Overall comment I'm not sure if I've misunderstood the findings but from my reading it appears that the approach the authors investigated was not a feasible one to address drug use and PTSD among SSW. This is not to say that the work is not valuable. Even showing that the approach was not feasible is an important advance in understanding how to target drug use and PTSD among this group. It may lead to more helpful ways that services might respond to these issues in a more sustainable way (outside of research funding and within service resources and capacity). There are also likely benefits that the women attained from engaging in this program. All of these issues could be incorporated into the discussion to provide different ways forward. Finally, did the authors include any feedback	This has been changed as requested (page 10, paragraph 2). This has been addressed as per reviewer 4's first comment. We hope this provides further clarity. This has been addressed as per reviewer 4's first comment. We hope this provides further clarity. While recognising that the reviewer had been understandably unclear about how many women had reached the final stage of the intervention, we still believe our statement to be true. We have tweaked it to reflect that it could be feasible and implemented with minor changes to the intervention (page 15, paragraph 2). We have made changes to the conclusions, in light of previous reviewers' comments with the fact that this was a feasibility study and not an effectiveness study. We have commented on the recruitment and retention of participants on page 15, last paragraph and page 16 first paragraph. We are cautious about drawing conclusions about how better services could support SSW as this study was not an effectiveness study – we do not know whether this intervention would yet effectively reduce drug and/or sex work from these data. The intervention was developed in response to evidence of a lack of consistent
--	---

mechanism of the study to the women in relation to their findings? If so could they include this in the paper.

CONSORT CHECKLIST

Generalisability – this is not addressed within the manuscript.

specialist trauma support for this group despite complex needs, which suggests the need to move away from informal support.

Table 1 represents the baseline data collected before and after informed consent to the study. We only collected PCL5 checklist scores for those participants whom provided informed consent at the point of the baseline data collection, so we have reported the number, percentage, median score and range in Table 1 for those consented participants.

For transparency purposes, we wanted to report on all participants in Table 3. We appreciate it may be easier to read by only reporting those that attended screening but it is clear from the 'attended trauma screening' column who did and did not attend.

We thank the reviewer for their commentary on our study and thorough review of it. We have highlighted that we made the difficult choice of including the most chaotic women who are most challenging to engage and who are often excluded from research studies. This was a calculated risk. We feel our results demonstrate evidence the intervention is feasible (four women reached the final stage), with the caveats for improvement for any future trial. Due to the relapsing and remitting nature of drug use and sex work and the far reaching effects they both have on the women's lives, there would be benefit to women less recently involved in sex work and drug use being in receipt of the intervention as our previous work demonstrates.⁶

This is the subject of a second paper as word limitations restrict us.

Yes we have fed back the results of the study to the three agencies we have worked with via a presentation, who in turn have fed this back to the women who attend the sex worker charity. We plan to produce posters and flyers to send distribute to sex workers who attend the charity,

	but due to journal copyright rules, we cannot do this until the results are published. As soon as the paper has been accepted for publication, this will be completed. We completed the STROBE checklist as this was the closest to a feasibility study. The study is not generalisable to the public and this should have been indicated better on the checklist as not applicable. We have updated the STROBE checklist accordingly.
--	--

VERSION 2 – REVIEW

REVIEWER	Alison Munro University of Dundee, Scotland, UK.
REVIEW RETURNED	25-Aug-2020

GENERAL COMMENTS	The comments that I made previously have all been addressed by the authors and I have no additional major comments to make. Thank you for addressing them. The paper, in my view, remains well written and presented and indeed with additions and edits has added yet more clarity to the paper. Abstract and Introduction Sections Both of these sections are comprehensive, succinct and explain the study as a whole, as well as set the scene for the relevance/importance of this feasibility study. Methods This section is very detailed and explains the methods clearly. Results The results are clearly presented and readily understood. Conclusions The conclusions are interesting, well presented and highlight the difficulties in recruiting and retaining, in research and in treatment, this group of women who use drugs.
---

REVIEWER	Amanda Roxburgh Burnet Institute, Australia
REVIEW RETURNED	13-Aug-2020

GENERAL COMMENTS	The paper is a little clearer on the outcomes and whether any of the participants finished the intervention. Some of my original comments have not been addressed. I include these along with other comments below. Abstract Objectives Can the authors please specify a little more clearly that the aims of the intervention were targeting both drug use and PTSD. This wasn't
--

	clear from the abstract. Introduction As per above comment, page 4 lines 46 and 47 – can the authors be more explicit here that the intervention is targeting drug use as well as trauma treatment. Methods PPI Can the authors describe how this co-design operated. How did they invite peers to be involved? How did they contribute to study design etc. A little more detail on the process of peer involvement would be useful. The intervention Page 5 - line 52 Sentence starting “The intervention consisted of SSW only drug treatment groups” Should this read “SSW drug treatment only groups”? Page 6 – line 7 “when group facilitators judged participants were achieving drug use stabilisation.” Can the authors describe what criteria were used to determine ‘stable drug use’? Data analysis Cost analysis Were staff costs factored in even when sessions didn’t go ahead due to non-attendance? Was this an expense for the study? If so, this should be articulated. Results Were any other demographic details collected on participants? Education? Housing etc? Group attendance Discussion Comparison with other research Page 15 lines 30 & 31 “ . . we showed how an integrated treatment approach in this complex vulnerable group can be feasibly implemented and delivered.” I’m still not sure the authors can make this statement as their results suggest that the treatment wasn’t feasibly implemented given low numbers going through to the final stages of the intervention in the context of high cost. In addition, there now seems to be a tension between the suggestion made here of “small changes made to the intervention”, and the suggestion made in the conclusions and implications to recruit more stable women. If the point of this intervention was to try and provide support and treatment for the women with the most complex of needs, I think the authors need to articulate how they might better do this for these groups of women in a more sustainable way. It may be perhaps that longer term responses are required in providing stability first for these women before moving into specialised trauma treatment. This requires more discussion. Conclusions and implications
--	---

	I think the authors could reflect in this section on how services might better support SSW in relation to their drug use and PTSD in more informal ways given that a more structured treatment program didn't seem feasible? Table 3 This table would be clearer if the authors highlighted only those women who attended trauma screening, in order to differentiate the 2 groups.
--	---

VERSION 2 – AUTHOR RESPONSE

Reviewer 4 comments	Responses
The paper is a little clearer on the outcomes and whether any of the participants finished the intervention. Some of my original comments have not been addressed. I include these along with other comments below.	We thank the reviewer for their comment that the paper is clearer. We had responded to all of reviewer 4's comments in the previous round of reviews. However, we have re-responded to the repeat comments and referenced our previous responses to these.
Abstract – Objectives: Can the authors please specify a little more clearly that the aims of the intervention were targeting both drug use and PTSD. This wasn't clear from the abstract.	Yes, we have added in the following words to the fourth sentence in the objectives (in bold): “.....to address post-traumatic stress disorder (PTSD) alongside drug treatment may therefore improve treatment outcomes.”
Introduction As per above comment, page 4 lines 46 and 47 – can the authors be more explicit here that the intervention is targeting drug use as well as trauma treatment.	Yes, we have changed the start of paragraph four to (in bold): “In collaboration with SSWs and service providers and informed by existing research [refs] we developed a novel intervention, to simultaneously address the unique and complex combination of drug use and PTSD in female drug-dependent SSWs.”
Methods PPI Can the authors describe how this co-design operated. How did they invite peers to be involved? How did they contribute to study design etc. A little more detail on the process of peer involvement would be useful.	Yes, as stated when we responded to the previous round of reviews (see *below), the PPI group was initially convened with the help of the sex worker charity. Staff at the sex worker charity, in contact with sex workers, informed them of the study set up and invited them to participate in the PPI group. Meetings were held before and during study set up, where all aspects of the study design were discussed with them including invitations/flyers and information sheets, the recruitment process, data collection processes, location and duration of drug groups and treatment, and many other aspects of the study. They recommended changes to aspects of

	documentation wording, text reminders and providing sandwiches. Several meetings were held during the study to improve aspects of the study, including recruitment, which is a normal practice for a feasibility study. We have modified paragraph 3 of the Methods (PPI section) to the following: “Women with experience of street sex work and drug-dependency took part in focus groups and one-to-one discussions with NJ to inform study design, processes, documentation and intervention development. On commencement of the study, a group of women who were ineligible for recruitment were approached for involvement in PPI. They addressed challenges with recruitment, participation and adherence issues (described below) and suggested solutions, which were implemented. For example, they recommended changes such as provision of sandwiches to improve attendance.” *Previous response to original comment: The PPI group was initially convened with the help of the sex worker charity. The PPI process is explained in the referenced published protocol paper (https://pubmed.ncbi.nlm.nih.gov/30391916/) which was referred to in the introduction and has been referred to more clearly at the beginning of the Methods section now (page 5, first paragraph). “Intervention and study design was developed based on input from the study PPI group that included women currently and previously involved in SSW and illicit drug use. The group convened before and during set up, contributed to the protocol development as well as the design of participant facing study documentation. Subsequent meetings have informed recruitment, topic guides, plain language study summary and plans for study dissemination. Ongoing PPI meetings will focus on troubleshooting issues identified during the study process and at the end of the study will focus on interpretation of results and dissemination methods.”
The intervention Page 5 - line 52 Sentence starting “The intervention consisted of SSW only drug treatment groups” Should this read “SSW drug	As we stated in the previous round of reviews, this was not the case. The emphasis is that the groups were SSWs only for all the reasons

treatment only groups”?	mentioned in the introduction. We also feel it is important to keep the terminology consistent with the published protocol paper.² *Previous response to original comment: We have kept the terminology consistent with the published protocol paper.
Page 6 – line 7 “when group facilitators judged participants were achieving drug use stabilisation.” Can the authors describe what criteria were used to determine ‘stable drug use’?	Yes, as we stated in the previous round of reviews (see *below), this was in the protocol paper. We have now added this into this paper - additional text has been added to the first paragraph of the Intervention section (fourth sentence): “As stated in the protocol, and in line with provider’s usual care, participants were perceived as demonstrating drug use stability by exhibiting evidence of life/drug use stability such as engagement and functioning in the group, positive interaction with group facilitators and regular opioid substitution therapy (OST) by the group facilitators.” *Previous response to original comment: The protocol paper provides more details about the methods as we could not recount all details of the methods fully due to word limitations. The protocol paper states that “After attending four group meetings, if participants are perceived by the group facilitators as exhibiting evidence of life/drug use stability such as engagement and functioning in the group, positive interaction with group facilitators, regular OST, they will be offered transfer to the ‘Preparation for Recovery’ group.”
Data analysis Cost analysis Were staff costs factored in even when sessions didn’t go ahead due to non-attendance? Was this an expense for the study? If so, this should be articulated.	Yes, we replied to this original comment in the previous round of reviews (see *below). A second paragraph has now been added to the ‘Economic data collection’ section of the Methods to clarify what happened for non-attended sessions as follows: “Non-attendance was dealt with as follows; if no participants arrived after 45 minutes for a ‘getting started’ session, staff left and were only costed for the time that they spent waiting. Staff delivering ‘trauma screening’, ‘1-1s’ and ‘stabilisation groups’ were costed for all sessions booked, regardless of non-attendances.”

	Non-attendance at drug groups, '1-1s' and 'stabilisation groups' were not a cost to this study, due to the generosity of the study partners. *Previous response to original comment: - Getting started: Staff waited for a maximum of 45 minutes to see if any attendees arrived. If no attendees arrived the staff left, to undertake other work and were thus only costed for the first section of the session. - Trauma screening and 1-1s: Staff costs were factored in even when sessions didn't go ahead (non-attendance) due to a contractual agreement with the psychologist . - Stabilisation group: No scheduled stabilisation groups had non-attendances.
Results Were any other demographic details collected on participants? Education? Housing etc? Group attendance	We replied to this comment in the previous round of reviews (see *below). No, we collected the minimum amount of data to avoid overburdening the participants with unnecessary information to assess the feasibility of the intervention. Basic demographic details were collected, but information on education and housing were unlikely to be helpful in assessing the outcome of this feasibility study. For a future effectiveness trial, this is important information, but as with all studies and trials, there is a balance required so as not to overburden participants with data collection versus being able to answer the questions posed by the study or trial. *Previous response to original comment: No, we collected only the data we required to answer the aims of the feasibility study and additional demographics beyond age, sex and permitted contact details were not required. If the intervention were to go to a full effectiveness trial, then collection of further demographic information may be ethical and warranted. We were mindful of the data burden to participants.
Discussion Comparison with other research Page 15 lines 30 & 31 “ . . . we showed how an integrated treatment approach in this complex vulnerable group can be feasibly implemented and delivered.”	We replied to this comment in the previous round of reviews (see *below). We had changed the sentence quoted by the reviewer to this in the last review; “...we showed how an integrated treatment

I'm still not sure the authors can make this statement as their results suggest that the treatment wasn't feasibly implemented given low numbers going through to the final stages of the intervention in the context of high cost.	approach in this complex vulnerable group could feasibly be implemented and delivered, with small changes to the intervention.” which doesn't appear to have been acknowledged in this round of comments. This was not a cost-effectiveness study and conclusions about context of high cost against numbers cannot accurately/scientifically be drawn using this study design. However, we have changed the sentence to acknowledge the high costs, and have removed the word 'small': “...we showed how an integrated treatment approach in this complex vulnerable group could feasibly be implemented and delivered, with changes to the intervention, albeit at a higher than expected cost, mostly due to the delays incurred due to service retendering.” We believe we successfully managed some women through the whole process of the intervention, which in our protocol paper was the aim of the feasibility study. We managed this, even though there were huge changes in services at the time (which led to problems) and we established during the process that without modification, the intervention was not appropriate for those women so actively involved in street sex work and illicit drug use. This was an innovative intervention in a group of very vulnerable women typically with high incidences of PTSD. We had acknowledged in the limitations that the costs are preliminary and that the potential to reduce costs cannot be examined. *Previous response to original comment: While recognising that the reviewer had been understandably unclear about how many women had reached the final stage of the intervention, we still believe our statement to be true. We have tweaked it to reflect that it could be feasible and implemented with minor changes to the intervention (page 15, paragraph 2).
In addition, there now seems to be a tension between the suggestion made here of “small changes made to the intervention”, and the suggestion made in the conclusions and implications to recruit more stable women. If the point of this intervention was to try and provide support and treatment for the women with the	The original aim of the intervention was hypothesized based on previous work, to attempt to support those women with the most complex needs. This was because (a) they are often neglected in services and research and (b) if these women were enabled to move through this intervention, then there was potential scope in the

most complex of needs, I think the authors need to articulate how they might better do this for these groups of women in a more sustainable way. It may be perhaps that longer term responses are required in providing stability first for these women before moving into specialised trauma treatment. This requires more discussion.	future for more stable women to be supported. We knew that achieving drug use stability in SSWs caught up in active sex working and drug dependency is virtually impossible, unless they reduce their drug use, which is why this intervention was created in the first place. Rather than launch into a full randomised controlled trial, with the assumption the intervention itself would work for the most challenging women, we conducted a feasibility study to explore if the intervention was do-able and acceptable to these women. As with most feasibility studies, changes to the intervention are usual. By suggesting changes to the inclusion criteria does not necessarily negate that women with more complex needs or chaotic lifestyles could not benefit from this intervention or would be excluded. Just that, more stable women could also benefit if they were included. Nevertheless, we understand the reviewer's comments in that we do not discuss this fully. To address this, we have added the following to the discussion and conclusion sections as follows: First paragraph of the discussion: "Managing SSW trauma disclosure proved challenging for drug group facilitators and non-clinical staff and resulted in the recommendation that there is additional training and support for staff in future studies. The need for intervention refinement, for example, extending drug stabilisation sessions, were suggested to provide additional support prior to trauma treatment." Second paragraph of the conclusions section: "They also suggested more staff support for managing trauma disclosure, extended drug stabilisation sessions and closer working could improve intervention delivery." Additional modifications to the recommendation points of the conclusion sections:  1. "The intervention could also include women with more stability in their lives to increase recruitment and retention. 6. An extended trauma therapy programme, including extended stabilisation therapy prior to trauma treatment, to accommodate the complexity of SSW needs
Conclusions and implications	

I think the authors could reflect in this section on how services might better support SSW in relation to their drug use and PTSD in more informal ways given that a more structured treatment program didn't seem feasible?	We replied to this comment in the previous round of reviews (see *below). Our intervention using specialist services was developed precisely because there is little or no evidence that more informal support from non-specialists works (see Roberts et al, 2016 systematic review).¹ We do not think speculating on this in this paper is possible, given the lack of evidence for them. We have made changes (as mentioned in the comment above) to say that those with very high complex needs may need additional support throughout the intervention. *Previous response to original comment: We are cautious about drawing conclusions about how better services could support SSW as this study was not an effectiveness study – we do not know whether this intervention would yet effectively reduce drug and/or sex work from these data. The intervention was developed in response to evidence of a lack of consistent specialist trauma support for this group despite complex needs, which suggests the need to move away from informal support.
Table 3 This table would be clearer if the authors highlighted only those women who attended trauma screening, in order to differentiate the 2 groups.	We replied to this comment in the previous round of reviews (see *below). However, we acknowledge the reviewer's comment about this again and have re-arranged the table so that those women who attended trauma screening are separated in the first four rows for clarity. We hope this is acceptable to the reviewer. We feel readers and other researchers would appreciate all data from all those involved in the intervention. (Please note that track-changes were not used on the contents of this table, as its display would be more complex to review.) *Previous response to original comment: For transparency purposes, we wanted to report on all participants in Table 3. We appreciate it may be easier to read by only reporting those that attended screening but it is clear from the 'attended trauma screening' column who did and did not attend.
Reviewer 1 comments	Responses
The comments that I made previously have all	We thank the reviewer for this comment and we

been addressed by the authors and I have no additional major comments to make. Thank you for addressing them.	are pleased that we addressed all their comments fully.
The paper, in my view, remains well written and presented and indeed with additions and edits has added yet more clarity to the paper.	We are grateful to the reviewer for this comment.
Abstract and Introduction Sections Both of these sections are comprehensive, succinct and explain the study as a whole, as well as set the scene for the relevance/importance of this feasibility study. Methods This section is very detailed and explains the methods clearly. Results The results are clearly presented and readily understood. Conclusions The conclusions are interesting, well presented and highlight the difficulties in recruiting and retaining, in research and in treatment, this group of women who use drugs	We thank the reviewer for all these remaining comments about our revisions.

References:

1. Roberts NP, Roberts PA, Jones N, et al. Psychological therapies for post-traumatic stress disorder and comorbid substance use disorder. Cochrane Database Syst Rev 2016;4(4):CD010204. doi: 10.1002/14651858.CD010204.pub2 [published Online First: 2016/04/05]
2. Jeal N, Patel R, Redmond NM, et al. Drug use in street sex workers (DUSK) study protocol: a feasibility and acceptability study of a complex intervention to reduce illicit drug use in drug-dependent female street sex workers. BMJ Open 2018;8(11):e022728. doi: 10.1136/bmjopen-2018-022728 [published Online First: 2018/11/06]

VERSION 3 – REVIEW

REVIEWER	Amanda Roxburgh Burnet Institute, Australia
REVIEW RETURNED	29-Oct-2020
GENERAL COMMENTS	Accept